# Trigonometric function transformation and its application in software reliability modeling

**Dai-Nghia Vy[1], Van-Thuan Nguyen[1]\*, Quyet-Thang Huynh[2],
Trung-Nghia Phung[3], Hung-Cuong Nguyen[1]**

1 Faculty of Engineering Technology, Hung Vuong University, Phu Tho, Vietnam, 2 Hanoi University of Science and Technology, Hanoi, Vietnam, 3 Thai Nguyen University of Information Communication Technology, Thai Nguyen, Vietnam

\* nguyenvanthuan@hvu.edu.vn

## Abstract

**Context:** Many models based on S-shaped functions demonstrate their advantages in non-homogeneous Poisson process software reliability modeling. However, three well-known types have been used without deep mathematical evaluation. Furthermore, some other promising S-shaped functions should be aimed at.

**Objectives:** (1) Transform the sine function to change the domain and co-domain without losing the S-shaped pattern, and (2) Evaluate four types of S-shaped functions mathematically, including three well-known types and this newly considered type of transformation form of the trigonometric function.

**Methods:** The transformation is taken by a four-step process, including horizontal and vertical shift, horizontal and vertical scale, to maintain the S-shaped form. The mathematical evaluation is performed using numerical analysis techniques in four aspects of function: special cases, domain, range, and limitations.

**Results:** The first contribution is a deep mathematical appreciation of three well-known S-shaped functions. The second is the mathematical transformation of the trigonometric function to meet the real assumption. The last are the advantages and the applicability of this transformation in software reliability modeling.

**Conclusion:** Besides three well-used S-shaped functions, the transformation of the trigonometric function has a new advantage. Most importantly, there is a possibility of using the trigonometric function to introduce a new software reliability model.

## 1 Introduction

Over seven decades of affecting all the areas of human life [1], the number of developed software has exceeded 10 million. Software engineering [2], including methodology, programming language, framework, and tool, becomes increasingly complex to meet real-life requirements. Those complicated problems led to the insistence on maintaining the software quality. In the scope of this paper, the regarded topic is non-homogeneous Poisson process (NHPP) software reliability modeling (SRM),

**Data availability statement:** All relevant data are within the paper.

**Funding:** This research is supported by Hung Vuong University under grant number HV33.2024. The funders had no role in study design, data collection and analysis, decision to publish, or preparation of the manuscript.

**Competing interests:** The authors have declared that no competing interests exist.

that is, the application of probability technique in evaluating one of software quality characteristics [3,4].

Among models belonging to the NHPP SRMs group [5], the S-shaped curves have been widely studied. This term *S-shaped* is based on the pattern of the letter S, the nineteenth letter of the Latin alphabet. The S-shaped curve increases continuously over time when the growth rate is time-dependent. Geometrically, an S-shaped curve is a merging of convex and concave parts. Some studied S-shaped functions [6–13] verify the usability of this shape. However, previous publications have not thoroughly reviewed those functions analytically. In addition, some well-known trigonometric functions contain some parts that are S-shaped. Notwithstanding, the domain and codomain of trigonometric functions are bounded, so it does not reflect real-life quantities. So this paper (1) deeply analyzes some used S-shaped functions to summarize the advantages and the flaws, and subsequently (2) discusses a new type of S-shaped function, the transformation of the trigonometric function.

The rest of this article is organized into the following sections. The second section discusses the literature on the S-shaped functions in NHPP SRM. The next section introduces the transformation of the trigonometric function. The third section analyses those four S-shaped functions mathematically. Section 3 evaluates the applicability of the transformation in NHPP SRM. The last section summarizes all works and proposes some promising works to extend.

## 2 Software reliability modelling

### 2.1 NHPP SRM

The software testing timeline is modeled using the Poisson process. The terminology *non-homogeneous* is reflected by the parameter of the Poisson distribution, which is a time-dependent function. By time $t$, the software system has $a(t)$ faults while testers detected and removed $m(t)$ of them. So there are still $a(t)$–$m(t)$ faults. Consider a short period $\Delta t$ from $t$ to $t + \Delta t$, the number of detected faults is $m(t)$ and $m(t + \Delta t)$, respectively. Therefore, this period detects $m(t+\Delta t) - m(t)$ faults. As $\Delta t$ is a short time, assuming that the number of remaining faults, i.e., $a(t)$–$m(t)$, and the fault detection rate, i.e., $b(t)$, are constant. With notation of $o(\cdot)$ which $\lim_{\Delta t \to 0} o(\Delta t) = 0$, there is a relationship between those quantities:

$$m(t + \Delta t) - m(t) = b(t)[a(t) - m(t)]\Delta t + o(\Delta t) \tag{1}$$

Let $\Delta t \to 0$, the differential equation is:

$$\frac{\partial}{\partial t}m(t) = b(t)[a(t) - m(t)] \tag{2}$$

The general solution [10] of Eq (2) is given by:

$$m(t) = e^{-B(t)}\left[m_0 + \int_{t_0}^{t} a(\tau)b(\tau)e^{B(\tau)}d\tau\right] \tag{3}$$

where

$$B(t) = \int_{t_0}^{t} b(\tau)d\tau \tag{4}$$

In most cases, researchers focus on $m(0) = 0$ and $t_0 = 0$, so (3) can be shortened to:

$$m(t) = e^{-B(t)} \int_{0}^{t} a(\tau)b(\tau)e^{B(\tau)}d\tau \tag{5}$$

where

$$B(t) = \int_{0}^{t} b(\tau)d\tau \tag{6}$$

The main obstacle to proposing NHPP SRM is the integral calculation in Eq (5) when specific functions $a(t)$ and $b(t)$ are applied.

## 2.2 S-shaped functions in SRM

The term *S-shaped* reflects the growth of the function and can be partitioned into 5 phases:

- *Phase 1*. Firstly, the function increases from a lower bound with a small growth rate.
- *Phase 2*. Next, it increases the growth rate; therefore, the function increases more quickly.
- *Phase 3*. The function goes through *inflection point*, which is the point that the first derivation of the function changes from positive to negative quantities.
- *Phase 4*. The growth rate decreases, so the function increases more slowly.
- *Phase 5*. Lastly, the function converges to the upper bound.

Generally, the 2nd and 4th phases are the main parts of the S-shaped function, which are the convex and concave functions, respectively. There are some types of S-shaped functions in NHPP SRM that have been used as fault detection rates. First form [6,7] is Eq (7):

$$b_1(t) = \frac{b^2 t}{bt + 1} \tag{7}$$

Substitute into Eq (5):

$$m_1(t) = \frac{bt + 1}{e^{bt}} \int_{0}^{t} a(\tau) \frac{b^2 \tau e^{b\tau}}{(b\tau + 1)^2} d\tau \tag{8}$$

Inflection form [8–11] is in Eq (11):

$$b_2(t) = b\frac{1}{1 + \beta e^{-bt}} \tag{9}$$

Substitute into Eq (5):

$$m_2(t) = \frac{b}{e^{bt} + \beta} \int_0^t a(\tau)e^{b\tau} d\tau \tag{10}$$

And extended inflection form [12–14] is in Eq (11):

$$b_3(t) = b\frac{1 + k\beta e^{-bt}}{1 + \beta e^{-bt}} \tag{11}$$

Substitute into Eq (5):

$$m_3(t) = \frac{b(e^{bt} + \beta)^{k-1}}{e^{bkt}} \int_0^t a(\tau)e^{bk\tau} \frac{(1 + k\beta e^{-b\tau})(e^{-b\tau})^{k-1}}{(1 + \beta e^{-b\tau})^k} d\tau \tag{12}$$

There are some studied models which use S-shaped functions in Table 1.

## 3 Trigonometric function transformation

### 3.1 Trigonometric function

Originally, trigonometric functions were the ratios between angles of the right triangle. The first *analysis* consideration [15] of a trigonometric function is in 1669 by the derivation of the power series for the sine function of Isaac Newton. There are three famous trigonometric functions sine $\sin(t)$, cosine $\cos(t)$, and tangent $\tan(t)$. Their reciprocals are the cosecant, the secant, and the cotangent, respectively. Each of those six functions has a corresponding inverse function (e.g. $\sin^{-1} t$ or $\sin^{-1}(t)$ or $\arcsin t$) and hyperbolic functions ($\sinh(t)$, $\cosh(t)$, $\tanh(t)$, $\coth(t)$, $\text{sech}(t)$, and $\text{csch}(t)$). The angle, i.e., the unit of the input of a trigonometric function, can be measured by degrees in geometry and radians in calculus, in which the value milestones are 360° and $2\pi$ and their multiples, respectively. Trigonometric functions are periodic functions which period is $2\pi$ or $\pi$, i.e. $\sin(t) = \sin(t + 2\pi) = \sin(t + 2k\pi)$ and $\tan(t) = \tan(t + \pi) = \tan(t + k\pi)$. The periodical of $\sin(t)$ is illustrated at the top of Fig 1.

The rest of this manuscript studies the most well-known trigonometric function $\sin(t)$. As shown at the bottom of Fig 1, a part of the $\sin(t)$ function in $[-\frac{\pi}{2}, \frac{\pi}{2}]$ has an S-shaped pattern. $\sin(t)$ increase in all of $t \in [-\frac{\pi}{2}, \frac{\pi}{2}]$ while growth rate in $[-\frac{\pi}{2}, 0)$ and $(0, \frac{\pi}{2}]$ are positive and negative, respectively.

### 3.2 Trigonometric function transformation

$\sin(t)$ has an S-shaped form, but the lower bound, upper bound, and growth rate are fixed [16]. In the period $[-\frac{\pi}{2}, \frac{\pi}{2}]$ of length $\pi$, the range of codomain is 2 when $\min \sin(t) = -1$ and $\max \sin(t) = 1$ and the inflection point is $(x, y) = (0, 0)$. To apply this trigonometric function, we need to transform it. Aspects that need to be transformed include (1) the domain range in which $\sin(t)$ remains an S-shaped pattern, (2) the codomain as a couple of (min, max), and (3) the position of

**Table 1**. Some existing NHPP SRMs.

| No | Model | Modeling functions |
|----|-------|--------------------|
| 1. | Inflection S-shaped [8] | $a(t) = a \quad b(t) = \dfrac{b}{1 + \beta e^{-bt}}$ <br><br> $m(t) = a \times \dfrac{e^{bt} - 1}{e^{bt} + \beta}$ |
| 2. | Delayed S-shaped [7] | $a(t) = a \quad b(t) = \dfrac{b^2 t}{bt + 1}$ <br><br> $m(t) = a[1 - (1 + bt)e^{-bt}]$ |
| 3. | PNZ [9] | $a(t) = a(1 + \alpha t) \quad b(t) = \dfrac{b}{1 + \beta e^{-bt}}$ <br> $m(t) = \dfrac{a}{1 + \beta e^{-bt}}[(1 - e^{-bt})(1 - \dfrac{\alpha}{b}) + \alpha t]$ |
| 4. | Pham exponential [10] | $a(t) = a e^{\alpha t} \quad b(t) = \dfrac{b}{1 + \beta e^{-bt}}$ <br><br> $m(t) = \dfrac{ab(e^{\alpha t} - e^{-bt})}{(\alpha + b)(1 + \beta e^{-bt})}$ |
| 5. | Pham Zhang [11] | $a(t) = a - k e^{-\alpha t} \quad b(t) = \dfrac{b}{1 + \beta e^{-bt}}$ <br> $m(t) = \dfrac{1}{1 + \beta e^{-bt}}[a(1 - e^{-bt}) - \dfrac{bk(e^{-\alpha t} - e^{-bt})}{b - \alpha}]$ |
| 6. | Pham parameter [6] | $a(t) = a(1 + bt)^2 \quad b(t) = \dfrac{b^2 t}{bt + 1}$ <br><br> $m(t) = a(bt + 1)(bt + e^{-bt} - 1)$ |
| 7. | Cuong-Thang 3-parameter S-shaped [14] [12] | $a(t) = a \quad b(t) = \dfrac{1 + k\beta e^{-bt}}{1 + \beta e^{-bt}}$ <br><br> $m(t) = a - \dfrac{a}{e^{bkt}}\left(\dfrac{e^{bt} + \beta}{1 + \beta}\right)^{k-1}$ |
| 8. | Imperfect debugging Cuong-Thang 3-parameter S-shaped [14] [12] | $a(t) = a \quad b(t) = \dfrac{1 + k\beta e^{-bt}}{1 + \beta e^{-bt}}$ <br><br> $m(t) = a - \dfrac{a}{e^{bkt}}\left(\dfrac{e^{bt} + \beta}{1 + \beta}\right)^{k-1}$ |

the inflection point as a moment when $\sin(t)$ changes its behavior. As illustrated in Fig 2, those aspects can be met by the mathematical transformation of $\sin(t)$, which includes 4 steps:

- *Step 1*. From $f(t) = \sin(t)$, it is scaled vertically to change the range of codomain, i.e., $\max \sin(t) - \min \sin(t)$:

$$f(t) = b \sin(t) \tag{13}$$

- *Step 2*. From Eq (13), it is scaled horizontally to change the length of the period in which $\sin(t)$ remains in an S-shaped pattern:

$$f(t) = b \sin(\beta t) \tag{14}$$

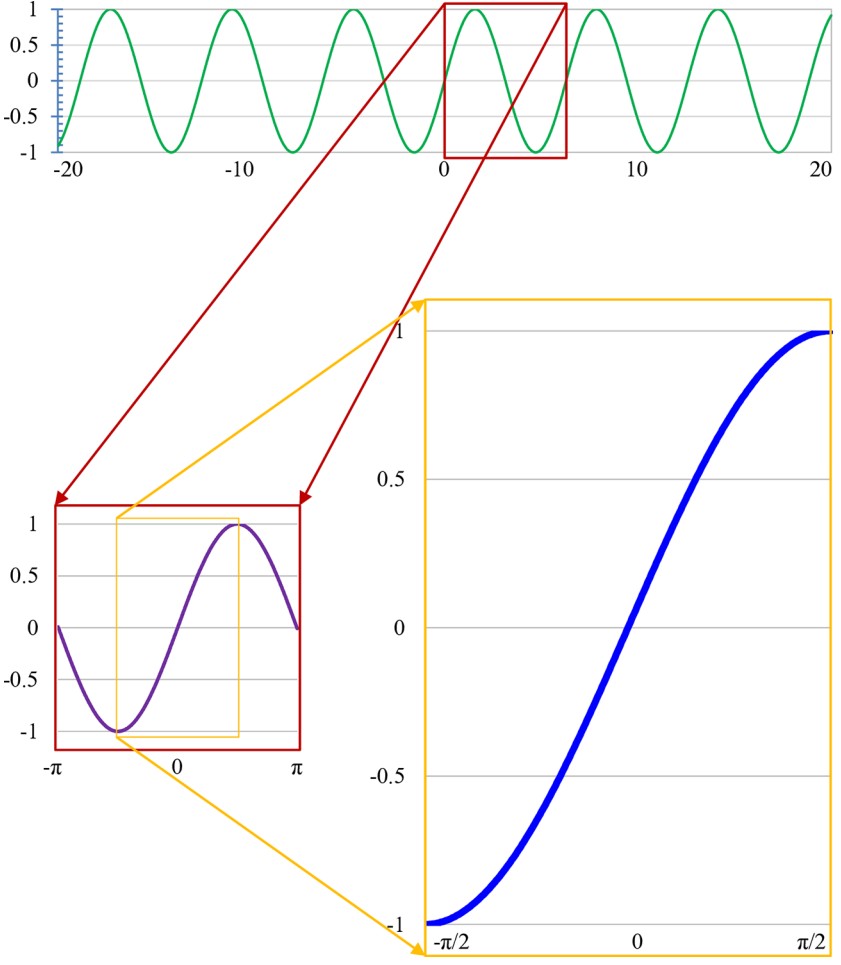

**Fig 1**. S-shaped as a part of sin($t$) function.

- *Step 3*. From Eq (14), it is shifted horizontally to change the bounds of the function domain:

$$f(t) = b \sin(\beta t + \varphi) \tag{15}$$

- *Step 4*. From Eq (15), it is shifted vertically to change the bounds of the function codomain:

$$b_4(t) = b \sin(\beta t + \varphi) + k \tag{16}$$

Table 2 shows the function $b_4(t)$ when some of variables vanishes. Because of their appearance in multiplication, $b$ and $\beta$ vanish when they take a value of 1. On the other hand, $\varphi$ and $k$ appear in addition, so they vanish when taking a value of 0. For example, when $\beta = 1$ and $\varphi = 0$, $b_4(t)$ becomes $b \sin(1.t + 0) + k = b_{4_9}(t) = b \sin(t) + k$.

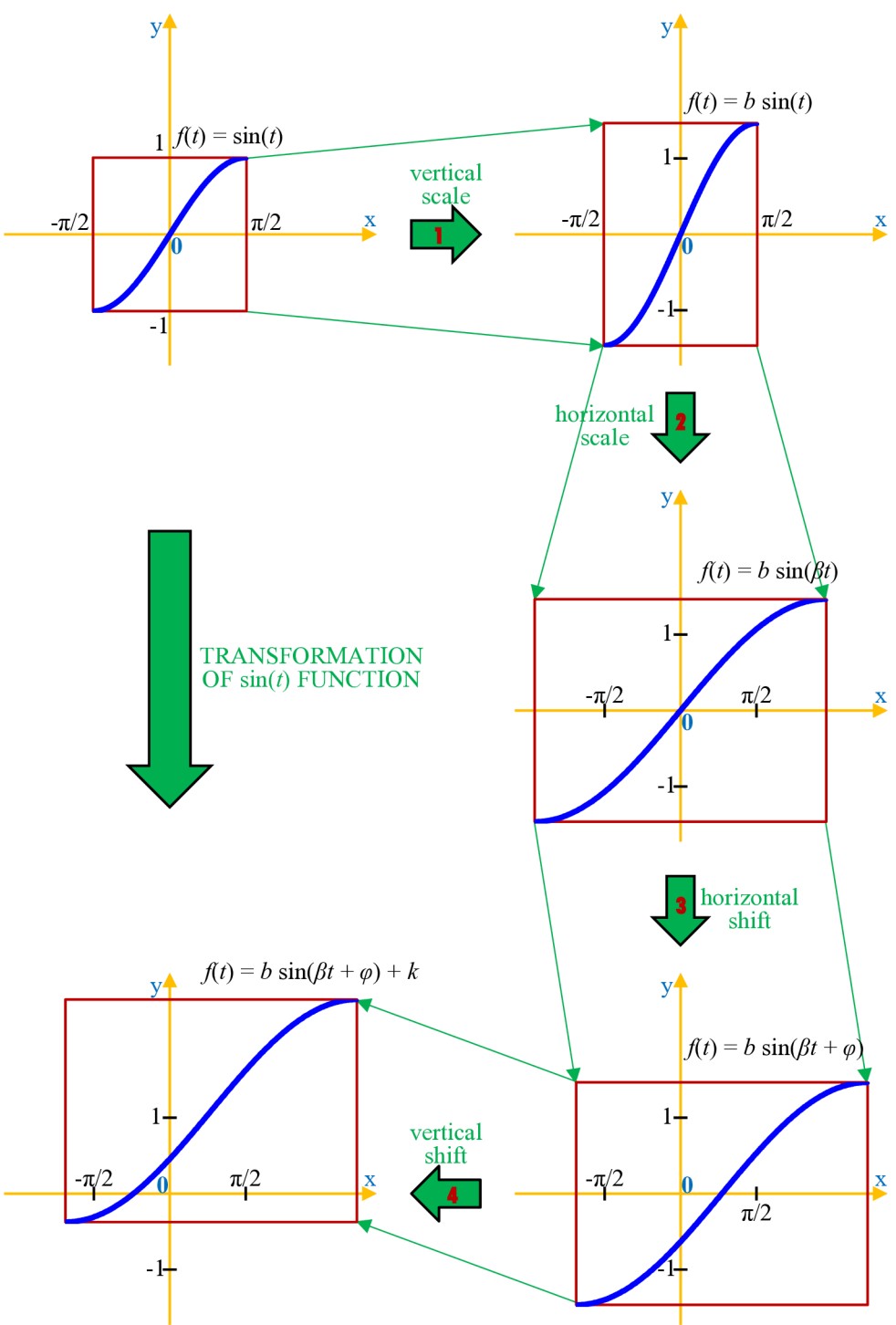

**Fig 2. Transformation from $f(t) = \sin(t)$ to $f(t) = b\sin(\beta t + \varphi) + k$.**

**Table 2. Function $b_4(t) = b\sin(\beta t + \varphi) + k$ when some variables are vanished.**

| Case | Vanishment | | | | $b_4(t)$ |
|---|---|---|---|---|---|
| | $b$ | $\beta$ | $\varphi$ | $k$ | |
| 1 | | | | | $b_{4_1}(t) = b\sin(\beta t + \varphi) + k$ <br><br> $m_{4_1}(t) = \dfrac{[e^{\cos(\beta t + \varphi)}]^{\frac{b}{\beta}}}{e^{kt}} \int\limits_0^t a(\tau)[b\sin(\beta\tau + \varphi) + k]\dfrac{e^{k\tau}}{[e^{\cos(\beta\tau+\varphi)}]^{\frac{b}{\beta}}}\,d\tau$ |
| 2 | ✓ | | | | $b_{4_2}(t) = \sin(\beta t + \varphi) + k$ <br><br> $m_{4_2}(t) = \dfrac{[e^{\cos(\beta t + \varphi)}]^{\frac{1}{\beta}}}{e^{kt}} \int\limits_0^t a(\tau)[\sin(\beta\tau + \varphi) + k]\dfrac{e^{k\tau}}{[e^{\cos(\beta\tau+\varphi)}]^{\frac{1}{\beta}}}\,d\tau$ |
| 3 | | ✓ | | | $b_{4_3}(t) = b\sin(t + \varphi) + k$ <br><br> $m_{4_3}(t) = \dfrac{[e^{\cos(t+\varphi)}]^b}{e^{kt}} \int\limits_0^t a(\tau)[b\sin(\tau + \varphi) + k]\dfrac{e^{k\tau}}{[e^{\cos(\tau+\varphi)}]^b}\,d\tau$ |
| 4 | | | ✓ | | $b_{4_4}(t) = b\sin(\beta t) + k$ <br><br> $m_{4_4}(t) = \dfrac{[e^{\cos(\beta t)}]^{\frac{b}{\beta}}}{e^{kt}} \int\limits_0^t a(\tau)[b\sin(\beta\tau) + k]\dfrac{e^{k\tau}}{[e^{\cos(\beta\tau)}]^{\frac{b}{\beta}}}\,d\tau$ |
| 5 | | | | ✓ | $b_{4_5}(t) = b\sin(\beta t + \varphi)$ <br><br> $m_{4_5}(t) = [e^{\cos(\beta t + \varphi)}]^{\frac{b}{\beta}} \int\limits_0^t a(\tau)[b\sin(\beta\tau + \varphi)]\dfrac{1}{[e^{\cos(\beta\tau+\varphi)}]^{\frac{b}{\beta}}}\,d\tau$ |
| 6 | ✓ | ✓ | | | $b_{4_6}(t) = \sin(t + \varphi) + k$ <br><br> $m_{4_6}(t) = \dfrac{e^{\cos(t+\varphi)}}{e^{kt}} \int\limits_0^t a(\tau)[\sin(\tau + \varphi) + k]\dfrac{e^{k\tau}}{e^{\cos(\tau+\varphi)}}\,d\tau$ |
| 7 | ✓ | | ✓ | | $b_{4_7}(t) = \sin(\beta t) + k$ <br><br> $m_{4_7}(t) = \dfrac{[e^{\cos(\beta t)}]^{\frac{1}{\beta}}}{e^{kt}} \int\limits_0^t a(\tau)[\sin(\beta\tau) + k]\dfrac{e^{k\tau}}{[e^{\cos(\beta\tau)}]^{\frac{1}{\beta}}}\,d\tau$ |
| 8 | ✓ | | | ✓ | $b_{4_8}(t) = \sin(\beta t + \varphi)$ <br><br> $m_{4_8}(t) = [e^{\cos(\beta t + \varphi)}]^{\frac{1}{\beta}} \int\limits_0^t a(\tau)[\sin(\beta\tau + \varphi)]\dfrac{1}{[e^{\cos(\beta\tau+\varphi)}]^{\frac{1}{\beta}}}\,d\tau$ |
| 9 | | ✓ | ✓ | | $b_{4_9}(t) = b\sin(t) + k$ <br><br> $m_{4_9}(t) = \dfrac{[e^{\cos(t)}]^b}{e^{kt}} \int\limits_0^t a(\tau)[b\sin(\tau) + k]\dfrac{e^{k\tau}}{[e^{\cos(\tau)}]^b}\,d\tau$ |
| 10 | | ✓ | | ✓ | $b_{4_{10}}(t) = b\sin(t + \varphi)$ <br><br> $m_{4_{10}}(t) = [e^{\cos(t+\varphi)}]^b \int\limits_0^t a(\tau)[b\sin(\tau + \varphi)]\dfrac{1}{[e^{\cos(\tau+\varphi)}]^b}\,d\tau$ |
| 11 | | | ✓ | ✓ | $b_{4_{11}}(t) = b\sin(\beta t)$ <br><br> $m_{4_{11}}(t) = [e^{\cos(\beta t)}]^{\frac{b}{\beta}} \int\limits_0^t a(\tau)[b\sin(\beta\tau)]\dfrac{1}{[e^{\cos(\beta\tau)}]^{\frac{b}{\beta}}}\,d\tau$ |
| 12 | ✓ | ✓ | ✓ | | $b_{4_{12}}(t) = \sin(t) + k$ <br><br> $m_{4_{12}}(t) = \dfrac{e^{\cos(t)}}{e^{kt}} \int\limits_0^t a(\tau)[\sin(\tau) + k]\dfrac{e^{k\tau}}{e^{\cos(\tau)}}\,d\tau$ |
| 13 | ✓ | ✓ | | ✓ | $b_{4_{13}}(t) = \sin(t + \varphi)$ <br><br> $m_{4_{13}}(t) = e^{\cos(t+\varphi)} \int\limits_0^t a(\tau)[\sin(\tau + \varphi)]\dfrac{1}{e^{\cos(\tau+\varphi)}}\,d\tau$ |
| 14 | ✓ | | ✓ | ✓ | $b_{4_{14}}(t) = \sin(\beta t)$ <br><br> $m_{4_{14}}(t) = [e^{\cos(\beta t)}]^{\frac{1}{\beta}} \int\limits_0^t a(\tau)[\sin(\beta\tau)]\dfrac{1}{[e^{\cos(\beta\tau)}]^{\frac{1}{\beta}}}\,d\tau$ |
| 15 | | ✓ | ✓ | ✓ | $b_{4_{15}}(t) = b\sin(t)$ <br><br> $m_{4_{15}}(t) = [e^{\cos(t)}]^b \int\limits_0^t a(\tau)[b\sin(\tau)]\dfrac{1}{[e^{\cos(\tau)}]^b}\,d\tau$ |
| 16 | ✓ | ✓ | ✓ | ✓ | $b_{4_{16}}(t) = \sin(t)$ <br><br> $m_{4_{16}}(t) = e^{\cos(t)} \int\limits_0^t a(\tau)\sin(\tau)\dfrac{1}{e^{\cos(\tau)}}\,d\tau$ |

### 3.3 Applying trigonometric function transformation in NHPP SRM

Substitute Eq (16) into (6), we have:

$$B(t) = \int_0^t \left[ b\sin(\beta\tau + \varphi) + k \right] d\tau = kt - \frac{b}{\beta}\cos(\beta t + \varphi) + \frac{b}{\beta}\cos(\varphi) \tag{17}$$

Eq (5) becomes:

$$m(t) = \frac{\left[ e^{\cos(\beta t + \varphi)} \right]^{\frac{b}{\beta}}}{e^{kt}} \int_0^t a(\tau) \left[ b\sin(\beta\tau + \varphi) + k \right] \frac{e^{k\tau}}{\left[ e^{\cos(\beta\tau + \varphi)} \right]^{\frac{b}{\beta}}} d\tau \tag{18}$$

To get a solution of Eq (18), we should choose one of the following ways:

- A suitable function $a(\tau)$.
- Considering the vanishment of some parameters of set $\{b, \beta, \varphi, k\}$.

or apply both on one single use. Substitute each case of $b_4(t)$ from Table 2 into Eq (18), $m_4(t)$ are in Table 2.

## 4 Numerical analysis

Let $Y(x)$ be a sign function of $x$. $\wedge$ is an AND logic operator.

### 4.1 First form

Let consider:

$$b_1(t) = \frac{b^2 t}{bt + 1} \tag{19}$$

If $b = 0$, $b_1(t)$ becomes a constant function of 0. If $b \neq 0$, we have:

$$\frac{\partial}{\partial t} b_1(t) = \frac{b^2}{(bt + 1)^2} \tag{20}$$

$$\frac{\partial^2}{\partial t^2} b_1(t) = \frac{-2b^3}{(bt + 1)^3} \tag{21}$$

The domain of $b_1(t)$, $\frac{\partial}{\partial t} b_1(t)$, and $\frac{\partial^2}{\partial t^2} b_1(t)$ are:

$$t \in \mathbb{R} \setminus \{\frac{-1}{b}\} \tag{22}$$

The range of $b_1(t)$, $\frac{\partial}{\partial t}b_1(t)$, and $\frac{\partial^2}{\partial t^2}b_1(t)$ are:

$$\mathrm{Y}\big(b_1(t)\big) = \mathrm{Y}\Big(-bt\big[\frac{-1}{b}-t\big]\Big) \tag{23}$$

$$\mathrm{Y}\Big(\frac{\partial}{\partial t}b_1(t)\Big) = \mathrm{Y}\big(\beta\big) \tag{24}$$

$$\mathrm{Y}\Big(\frac{\partial^2}{\partial t^2}b_1(t)\Big) = \mathrm{Y}\Big(\frac{-1}{b}-t\Big) \tag{25}$$

Fig 3 illustrates $b_1(t)$ in some cases of parameters: green line in case of $b = 1.5$, light blue line in case of $b = 0.5$, red line in case of $b = 1.0$, and orange line in case of $b = -1.0$. Pure lines are in rational cases and dashed lines in others.

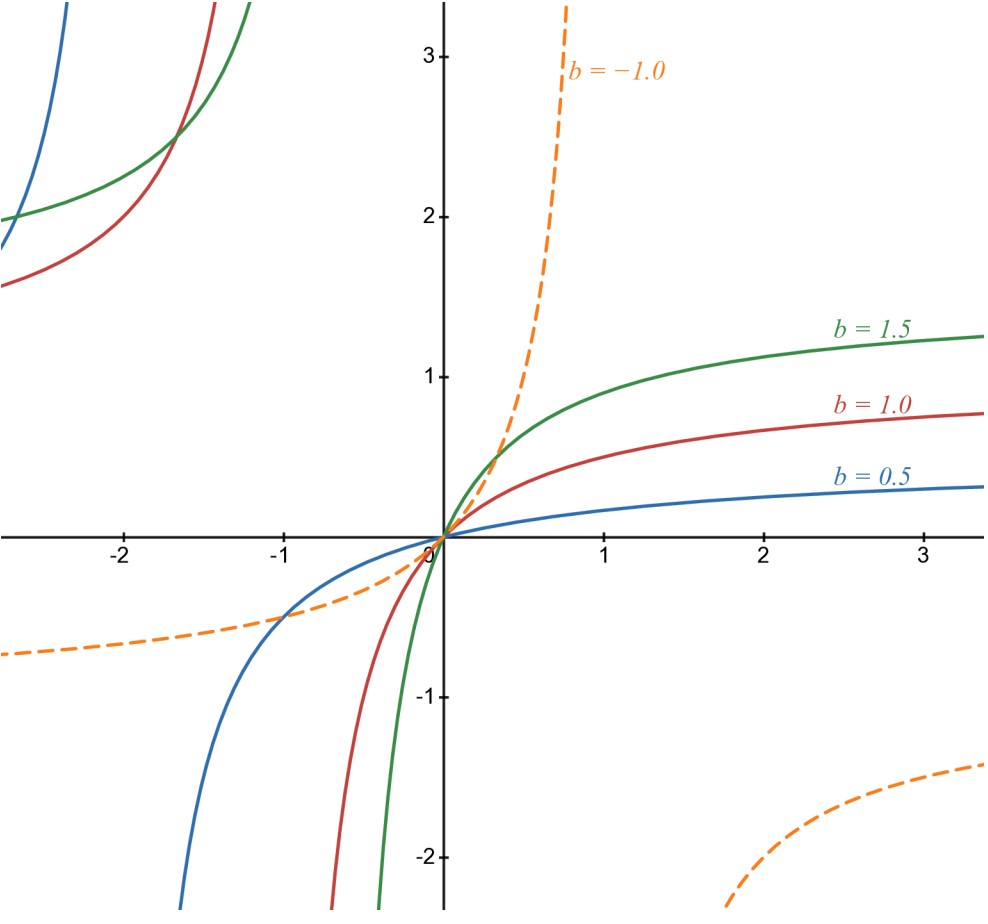

**Fig 3. Numerical illustration of $b_1(t)$.**

Limitations of $b_1(t)$ and $\frac{\partial}{\partial t}b_1(t)$ are in Table 3. So variation table of $b_1(t)$ when $b > 0$ and $b < 0$ are in Table 4 and Table 5, respectively. We can see that this form is not *fully S-shaped* because it only contains the two last phases of the S-shaped curve as described in sub-section 2.2. $b_1(t)$ does not have an early, quick growth period and an inflection point. In $[\frac{-1}{b}, +\infty)$, it grows slowlier because the $\frac{\partial^2}{\partial t^2}b_1(t)$ is negative, then $\frac{\partial}{\partial t}b_1(t)$ is decreased. Furthermore, $\lim_{t\to+\infty} b_1(t) = b$ reflects the convergence of this function to an upper bound as illustrated in Fig 3.

## 4.2 Second form

Let consider:

$$b_2(t) = b\frac{1}{1 + \beta e^{-bt}} \tag{26}$$

We have some special cases:

**Table 3. Limitations of $b_1(t)$ and $\frac{\partial}{\partial t}b_1(t)$.**

| $t \to$ | $-\infty$ | $+\infty$ | $\left(\frac{-1}{b}\right)^-$ | $\left(\frac{-1}{b}\right)^+$ |
|---|---|---|---|---|
| $b_1(t) \to$ | $b^+$ | $b^-$ | $+\infty$ | $-\infty$ |
| $\frac{\partial}{\partial t}b_1(t) \to$ | $0^+$ | $0^+$ | $+\infty$ | $+\infty$ |

**Table 4. Variation table of $b_1(t)$ when $b > 0$.**

| $t$ | $-\infty$ | | $\frac{-1}{b}$ | | $0$ | | $+\infty$ |
|---|---|---|---|---|---|---|---|
| $\frac{\partial^2}{\partial t^2}b_1(t)$ | | $+$ | | | | $-$ | |
| $\frac{\partial}{\partial t}b_1(t)$ | $0$ | $\nearrow$ | $+\infty$ / $+\infty$ | | | $\searrow$ | $0$ |
| $b_1(t)$ | $b$ | $\nearrow$ | $+\infty$ / $+\infty$ | $-\infty$ | $\nearrow$ | $0$ / $\nearrow$ | $b$ |

**Table 5. Variation table of $b_1(t)$ when $b < 0$.**

| $t$ | $-\infty$ | | $0$ | | $\frac{-1}{b}$ | | $+\infty$ |
|---|---|---|---|---|---|---|---|
| $\frac{\partial^2}{\partial t^2}b_1(t)$ | | | $+$ | | | $-$ | |
| $\frac{\partial}{\partial t}b_1(t)$ | $0$ | $\nearrow$ | | $+\infty$ / $+\infty$ | | $\searrow$ | $0$ |
| $b_1(t)$ | $b$ | $\nearrow$ | $0$ / $\nearrow$ | $+\infty$ / $+\infty$ | | $-\infty$ / $\nearrow$ | $b$ |

- If $(b, \beta) = (0, -1)$, $b_2(t)$ is indeterminate.
- If $b = 0$ and $\beta \neq -1$, $b_2(t)$ becomes constant function of 0.
- If $\beta = 0$, $b_2(t)$ becomes constant function of $b$.

In other cases in next subsub-sections when $b \neq 0$ and $\beta \neq 0$, we have:

$$\frac{\partial}{\partial t} b_2(t) = \frac{b^2 \beta e^{bt}}{(e^{bt} + \beta)^2} = \frac{b^2 \beta e^{-bt}}{(1 + \beta e^{-bt})^2} \tag{27}$$

$$\frac{\partial^2}{\partial t^2} b_2(t) = \frac{b^3 \beta e^{bt}(\beta - e^{bt})}{(e^{bt} + \beta)^3} = \frac{b^3 \beta e^{-bt}(\beta e^{-bt} - 1)}{(1 + \beta e^{-bt})^3} \tag{28}$$

The domain of $b_2(t)$, $\frac{\partial}{\partial t} b_2(t)$, and $\frac{\partial^2}{\partial t^2} b_2(t)$ are:

$$t \in \begin{cases} \mathbb{R} & \text{if } \beta > 0 \\ \mathbb{R} \setminus \{\frac{\ln(-\beta)}{b}\} & \text{if } \beta < 0 \end{cases} \tag{29}$$

The range of $b_2(t)$, $\frac{\partial}{\partial t} b_2(t)$, and $\frac{\partial^2}{\partial t^2} b_2(t)$ are:

$$\mathrm{Y}\big(b_2(t)\big) = \begin{cases} \mathrm{Y}\big(b\big) & \text{if } \beta > 0 \\ \mathrm{Y}\big(t - \frac{\ln |\beta|}{b}\big) & \text{if } \beta < 0 \end{cases} \tag{30}$$

$$\mathrm{Y}\Big(\frac{\partial}{\partial t} b_2(t)\Big) = \mathrm{Y}\big(\beta\big) \tag{31}$$

$$\mathrm{Y}\Big(\frac{\partial^2}{\partial t^2} b_2(t)\Big) = \mathrm{Y}\Big(\frac{\ln |\beta|}{b} - t\Big) \tag{32}$$

$\frac{\partial^2}{\partial t^2} b_2(t) = 0 \Leftrightarrow \beta > 0$ and $x_2 = \frac{\ln \beta}{b}$, then inflection point of $b_2(t)$ is $(x_2, y_2) = \big(\frac{\ln \beta}{b}, \frac{b}{2}\big)$ when $\beta > 0$. Limitations of $b_2(t)$, $\frac{\partial}{\partial t} b_2(t)$, and $\frac{\partial^2}{\partial t^2} b_2(t)$ are in Table 6.

So variation table of $b_2(t)$ are in Tables 7, 8, 9, and 10. Fig 4 illustrates $b_2(t)$ in some cases of parameters: blue line in case of $(b, \beta) = (3, 2)$, red line in case of $(b, \beta) = (2, 3)$, green line in case of $(b, \beta) = (1.2, -2.2)$, orange line in case of $(b, \beta) = (-1.4, 2.4)$, and purple line in case of $(b, \beta) = (-1.6, -2.6)$. Pure lines are in rational cases and dashed lines in others. The triangle with the corresponding filled color is the inflection point of each curve.

### 4.3 Third form

Let consider:

$$b_3(t) = b \frac{1 + k\beta e^{-bt}}{1 + \beta e^{-bt}} \tag{33}$$

**Table 6. Limitations of $b_2(t)$ and $\frac{\partial}{\partial t}b_2(t)$.**

| | $t \to$ | $-\infty$ | $+\infty$ | $\left(\frac{\ln|\beta|}{b}\right)^-$ | $\left(\frac{\ln|\beta|}{b}\right)^+$ |
|---|---|---|---|---|---|
| $b > 0, \beta > 0$ | $b_2(t) \to$ | $0^+$ | $b^-$ | $\left(\frac{b}{2}\right)^-$ | $\left(\frac{b}{2}\right)^+$ |
| | $\frac{\partial}{\partial t}b_2(t) \to$ | $0^+$ | $0^+$ | $\left(\frac{b^2}{4}\right)^-$ | $\left(\frac{b^2}{4}\right)^-$ |
| $b > 0, \beta < 0$ | $b_2(t) \to$ | $0^-$ | $b^+$ | $-\infty$ | $+\infty$ |
| | $\frac{\partial}{\partial t}b_2(t) \to$ | $0^-$ | $0^-$ | $-\infty$ | $-\infty$ |
| $b < 0, \beta > 0$ | $b_2(t) \to$ | $b^+$ | $0^-$ | $\left(\frac{b}{2}\right)^-$ | $\left(\frac{b}{2}\right)^+$ |
| | $\frac{\partial}{\partial t}b_2(t) \to$ | $0^+$ | $0^+$ | $\left(\frac{b^2}{4}\right)^-$ | $\left(\frac{b^2}{4}\right)^-$ |
| $b < 0, \beta < 0$ | $b_2(t) \to$ | $b^-$ | $0^+$ | $-\infty$ | $+\infty$ |
| | $\frac{\partial}{\partial t}b_2(t) \to$ | $0^-$ | $0^-$ | $-\infty$ | $-\infty$ |

**Table 7. Variation table of $b_2(t)$ when b > 0 and $\beta$ > 0.**

| $t$ | $-\infty$ | | $\frac{\ln\beta}{b}$ | | $+\infty$ |
|---|---|---|---|---|---|
| $\frac{\partial^2}{\partial t^2}b_2(t)$ | | $+$ | $0$ | $-$ | |
| $\frac{\partial}{\partial t}b_2(t)$ | $0$ | $\nearrow$ | $\frac{b^2}{4}$ | $\searrow$ | $0$ |
| $b_2(t)$ | $0$ | $\nearrow$ | $\frac{b}{2}$ | $\nearrow$ | $b$ |

**Table 8. Variation table of $b_2(t)$ when b > 0 and $\beta$ < 0.**

| $t$ | $-\infty$ | | $\frac{\ln(-\beta)}{b}$ | | $+\infty$ | |
|---|---|---|---|---|---|---|
| $\frac{\partial^2}{\partial t^2}b_2(t)$ | | $-$ | | | $+$ | |
| $\frac{\partial}{\partial t}b_2(t)$ | $0$ | $\searrow$ $-\infty$ | | $-\infty$ | $\nearrow$ | $0$ |
| $b_2(t)$ | $0$ | $\searrow$ $-\infty$ | | $+\infty$ | $\searrow$ | $b$ |

We have some special cases:

- If $(b, \beta) = (0, -1)$, $b_2(t)$ is indeterminate.
- If $b = 0$ and $\beta \neq -1$, $b_3(t)$ becomes constant function of 0.
- If $\beta = 0$, $b_3(t)$ becomes constant function of $b$.
- If $k = 1$, $b_3(t)$ becomes constant function of $b$.
- If $k = 0$, $b_3(t)$ becomes $b_2(t)$.

**Table 9**. Variation table of $b_2(t)$ when $b < 0$ and $\beta > 0$.

| $t$ | $-\infty$ | | $\frac{\ln \beta}{b}$ | | $+\infty$ |
|---|---|---|---|---|---|
| $\frac{\partial^2}{\partial t^2} b_2(t)$ | | $+$ | $0$ | $-$ | |
| $\frac{\partial}{\partial t} b_2(t)$ | $0$ | $\nearrow$ | $\frac{b^2}{4}$ | $\searrow$ | $0$ |
| $b_2(t)$ | $b$ | $\nearrow$ | $\frac{b}{2}$ | $\nearrow$ | $0$ |

**Table 10**. Variation table of $b_2(t)$ when $b < 0$ and $\beta < 0$.

| $t$ | $-\infty$ | | $\frac{\ln(-\beta)}{b}$ | | $+\infty$ | |
|---|---|---|---|---|---|---|
| $\frac{\partial^2}{\partial t^2} b_2(t)$ | | $-$ | | | $+$ | |
| $\frac{\partial}{\partial t} b_2(t)$ | $0$ | $\searrow$ | $-\infty$ | $-\infty$ | $\nearrow$ | $0$ |
| $b_2(t)$ | $b$ | $\searrow$ $-\infty$ | | $+\infty$ | $\searrow$ | $0$ |

In other cases in next subsub-sections when $b \neq 0$, $\beta \neq 0$, and $k \neq 1$, we have:

$$\frac{\partial}{\partial t} b_3(t) = -b^2 \frac{\beta(k-1)e^{bt}}{(e^{bt}+\beta)^2} = -b^2 \frac{\beta(k-1)e^{-bt}}{(1+\beta e^{-bt})^2} \tag{34}$$

$$\frac{\partial^2}{\partial t^2} b_3(t) = b^3 \frac{\beta(k-1)e^{bt}(e^{bt}-\beta)}{(e^{bt}+\beta)^3} = b^3 \frac{\beta(k-1)e^{-bt}(1-\beta e^{-bt})}{(1+\beta e^{-bt})^3} \tag{35}$$

The domain of $b_3(t)$, $\frac{\partial}{\partial t} b_3(t)$, and $\frac{\partial^2}{\partial t^2} b_3(t)$ are:

$$t \in \begin{cases} \mathbb{R} & \text{if } \beta > 0 \\ \mathbb{R} \setminus \{\frac{\ln(-\beta)}{b}\} & \text{if } \beta < 0 \end{cases} \tag{36}$$

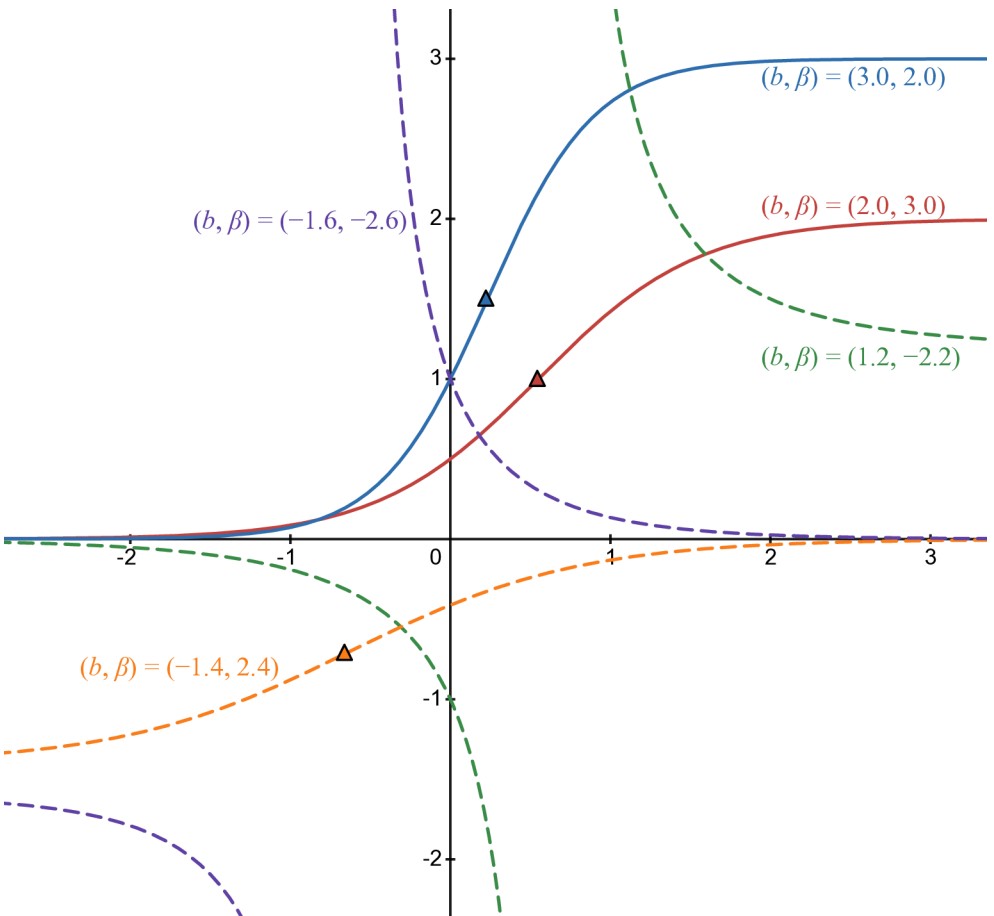

**Fig 4**. **Numerical illustration of $b_2(t)$.**

The range of $b_3(t)$, $\frac{\partial}{\partial t}b_3(t)$, and $\frac{\partial^2}{\partial t^2}b_3(t)$ are:

$$\mathrm{Y}\big(b_3(t)\big) = \begin{cases} \mathrm{Y}\big(b\big) & \text{if } \beta > 0, k > 0 \\ \mathrm{Y}\big(t - \frac{\ln|k\beta|}{b}\big) & \text{if } \beta > 0, k < 0 \\ \mathrm{Y}\big(b[t - \frac{\ln|k\beta|}{b}][t - \frac{\ln|\beta|}{b}]\big) & \text{if } \beta < 0, k > 0 \\ \mathrm{Y}\big(t - \frac{\ln|\beta|}{b}\big) & \text{if } \beta < 0, k < 0 \end{cases} \tag{37}$$

$$\mathrm{Y}\Big(\frac{\partial}{\partial t}b_3(t)\Big) = \mathrm{Y}\big(\beta(1 - k)\big) \tag{38}$$

$$\mathrm{Y}\Big(\frac{\partial^2}{\partial t^2}b_3(t)\Big) = \mathrm{Y}\big(b(1 - k)[\frac{\ln|\beta|}{b} - t]\big) \tag{39}$$

So $\frac{\partial^2}{\partial t^2} b_3(t) = 0 \Leftrightarrow \beta > 0$ and $x_3 = \frac{\ln \beta}{b}$, then inflection point of $b_3(t)$ is $(x_3, y_3) = \left(\frac{\ln \beta}{b}, \frac{b(k+1)}{2}\right)$ when $\beta > 0$. Table 11 are limitations of $b_3(t)$, $\frac{\partial}{\partial t} b_3(t)$, and $\frac{\partial^2}{\partial t^2} b_3(t)$. So variation table of $b_3(t)$ when $b > 0$, $\beta > 0$, and $k>1$ are in Tables 12, 13, 14, 15, 16, 17, 18, 19, 20, 21, 22, 23, 24, and 25.

Figs 5, 6, and 7 illustrate $b_3(t)$ in cases of parameters. Fig 5 illustrates when $b > 0$ and $\beta > 0$ which are green line in case of $(b, \beta, k) = (2, 1, -0.5)$, blue line in case of $(b, \beta, k) = (1.5, 2, 0.2)$, red line in case of $(b, \beta, k) = (1, 4, 2)$, and orange line in case of $(b, \beta, k) = (2, 1, -1.2)$.

Fig 6 corresponds with $b < 0$ and $\beta > 0$. The parameters are similar to Fig 5 except $b$ is negative: green line in case of $(b, \beta, k) = (-2, 1, -0.5)$, blue line in case of $(b, \beta, k) = (-1.5, 2, 0.2)$, red line in case of $(b, \beta, k) = (-1, 4, 2)$, and orange line in case of $(b, \beta, k) = (-2, 1, -1.2)$.

The last illustration of this third form is Fig 7 when $\beta < 0$: green line in case of $(b, \beta, k) = (1.5, -2, -1)$, blue line in case of $(b, \beta, k) = (1.5, -2, 0.5)$, red line in case of $(b, \beta, k) = (1, -4, 1.5)$, orange line in case of $(b, \beta, k) = (-1, -4, 1.5)$, purple

**Table 11. Limitations of $b_3(t)$ and $\frac{\partial}{\partial t} b_3(t)$.**

| | $t \to$ | $-\infty$ | $+\infty$ | $\left(\frac{\ln|\beta|}{|b|}\right)^-$ | $\left(\frac{\ln|\beta|}{|b|}\right)^+$ |
|---|---|---|---|---|---|
| $b > 0$ and $\beta > 0$ and $k>1$ | $b_3(t) \to$ | $bk^-$ | $b^+$ | $\left(\frac{b(k+1)}{2}\right)^+$ | $\left(\frac{b(k+1)}{2}\right)^-$ |
| | $\frac{\partial}{\partial t} b_3(t) \to$ | $0^-$ | $0^-$ | $\left(\frac{-b^2(k-1)}{4}\right)^+$ | $\left(\frac{-b^2(k-1)}{4}\right)^+$ |
| $b > 0$ and $\beta > 0$ and $k<1$ | $b_3(t) \to$ | $bk^+$ | $b^-$ | $\left(\frac{b(k+1)}{2}\right)^-$ | $\left(\frac{b(k+1)}{2}\right)^+$ |
| | $\frac{\partial}{\partial t} b_3(t) \to$ | $0^+$ | $0^+$ | $\left(\frac{-b^2(k-1)}{4}\right)^-$ | $\left(\frac{-b^2(k-1)}{4}\right)^-$ |
| $b > 0$ and $\beta < 0$ and $k>1$ | $b_3(t) \to$ | $bk^+$ | $b^-$ | $+\infty$ | $-\infty$ |
| | $\frac{\partial}{\partial t} b_3(t) \to$ | $0^+$ | $0^+$ | $+\infty$ | $+\infty$ |
| $b > 0$ and $\beta < 0$ and $k<1$ | $b_3(t) \to$ | $bk^-$ | $b^+$ | $-\infty$ | $+\infty$ |
| | $\frac{\partial}{\partial t} b_3(t) \to$ | $0^-$ | $0^-$ | $-\infty$ | $-\infty$ |
| $b < 0$ and $\beta > 0$ and $k>1$ | $b_3(t) \to$ | $b^-$ | $bk^+$ | $\left(\frac{b(k+1)}{2}\right)^+$ | $\left(\frac{b(k+1)}{2}\right)^-$ |
| | $\frac{\partial}{\partial t} b_3(t) \to$ | $0^-$ | $0^-$ | $\left(\frac{-b^2(k-1)}{4}\right)^+$ | $\left(\frac{-b^2(k-1)}{4}\right)^+$ |
| $b < 0$ and $\beta > 0$ and $k<1$ | $b_3(t) \to$ | $b^+$ | $bk^-$ | $\left(\frac{b(k+1)}{2}\right)^-$ | $\left(\frac{b(k+1)}{2}\right)^+$ |
| | $\frac{\partial}{\partial t} b_3(t) \to$ | $0^+$ | $0^+$ | $\left(\frac{-b^2(k-1)}{4}\right)^-$ | $\left(\frac{-b^2(k-1)}{4}\right)^-$ |
| $b < 0$ and $\beta < 0$ and $k>1$ | $b_3(t) \to$ | $b^+$ | $bk^-$ | $+\infty$ | $-\infty$ |
| | $\frac{\partial}{\partial t} b_3(t) \to$ | $0^+$ | $0^+$ | $+\infty$ | $+\infty$ |
| $b < 0$ and $\beta < 0$ and $k<1$ | $b_3(t) \to$ | $b^-$ | $bk^+$ | $-\infty$ | $+\infty$ |
| | $\frac{\partial}{\partial t} b_3(t) \to$ | $0^-$ | $0^-$ | $-\infty$ | $-\infty$ |

**Table 12. Variation table of $b_3(t)$ when $b > 0$, $\beta > 0$, and $k > 1$.**

| $t$ | $-\infty$ | | $\frac{\ln \beta}{b}$ | | $+\infty$ |
|---|---|---|---|---|---|
| $\frac{\partial^2}{\partial t^2} b_3(t)$ | | $-$ | $0$ | $+$ | |
| $\frac{\partial}{\partial t} b_3(t)$ | $0$ $\searrow$ | | $\frac{-b^2(k-1)}{4}$ | $\nearrow$ | $0$ |
| $b_3(t)$ | $bk$ $\searrow$ | | $\frac{b(k+1)}{2}$ | $\searrow$ | $b$ |

## Table 13. Variation table of $b_3(t)$ when $b > 0$, $\beta > 0$, and $0 < k < 1$.

| $t$ | $-\infty$ | | $\frac{\ln \beta}{b}$ | | $+\infty$ |
|---|---|---|---|---|---|
| $\frac{\partial^2}{\partial t^2} b_3(t)$ | | $+$ | $0$ | $-$ | |
| $\frac{\partial}{\partial t} b_3(t)$ | $0$ | $\nearrow$ | $\frac{-b^2(k-1)}{4}$ | $\searrow$ | $0$ |
| $b_3(t)$ | $bk$ | $\nearrow$ | $\frac{b(k+1)}{2}$ | $\nearrow$ | $b$ |

## Table 14. Variation table of $b_3(t)$ when $b > 0$, $\beta > 0$, and $-1 < k < 0$.

| $t$ | $-\infty$ | | $\frac{\ln(-k\beta)}{b}$ | | $\frac{\ln \beta}{b}$ | | $+\infty$ |
|---|---|---|---|---|---|---|---|
| $\frac{\partial^2}{\partial t^2} b_3(t)$ | | | $+$ | | $0$ | $-$ | |
| $\frac{\partial}{\partial t} b_3(t)$ | $0$ | $\nearrow$ | | | $\frac{-b^2(k-1)}{4}$ | $\searrow$ | $0$ |
| $b_3(t)$ | $bk$ | $\nearrow$ | $0$ | $\nearrow$ | $\frac{b(k+1)}{2}$ | $\nearrow$ | $b$ |

## Table 15. Variation table of $b_3(t)$ when $b > 0$, $\beta > 0$, and $k < -1$.

| $t$ | $-\infty$ | | $\frac{\ln \beta}{b}$ | | $\frac{\ln(-k\beta)}{b}$ | | $+\infty$ |
|---|---|---|---|---|---|---|---|
| $\frac{\partial^2}{\partial t^2} b_3(t)$ | | $+$ | $0$ | | $-$ | | |
| $\frac{\partial}{\partial t} b_3(t)$ | $0$ | $\nearrow$ | $\frac{-b^2(k-1)}{4}$ | | $\searrow$ | | $0$ |
| $b_3(t)$ | $bk$ | $\nearrow$ | $\frac{b(k+1)}{2}$ | $\nearrow$ | $0$ | $\nearrow$ | $b$ |

line in case of $(b, \beta, k) = (-1.5, -2, 0.5)$, and black line in case of $(b, \beta, k) = (-1.5, -2, -1)$. The triangle with the corresponding filled color is the inflection point of each curve. Pure lines are in rational cases and dashed lines in others.

**Table 16. Variation table of $b_3(t)$ when $b > 0$, $\beta < 0$, and $k > 1$.**

| $t$ | $-\infty$ | | $\frac{\ln(-\beta)}{b}$ | | $\frac{\ln(-k\beta)}{b}$ | | $+\infty$ | |
|---|---|---|---|---|---|---|---|---|
| $\frac{\partial^2}{\partial t^2}b_3(t)$ | | + | | | | − | | |
| $\frac{\partial}{\partial t}b_3(t)$ | 0 | ↗ | $+\infty$ | $+\infty$ | | ↘ | | 0 |
| $b_3(t)$ | $bk$ | ↗ | $+\infty$ | $-\infty$ | ↗ | 0 | ↗ | $b$ |

**Table 17. Variation table of $b_3(t)$ when $b > 0$, $\beta < 0$, and $0 < k < 1$.**

| $t$ | $-\infty$ | | $\frac{\ln(-k\beta)}{b}$ | | $\frac{\ln(-\beta)}{b}$ | | $+\infty$ | |
|---|---|---|---|---|---|---|---|---|
| $\frac{\partial^2}{\partial t^2}b_3(t)$ | | − | | | | + | | |
| $\frac{\partial}{\partial t}b_3(t)$ | 0 | ↘ | | $-\infty$ | $-\infty$ | ↗ | | 0 |
| $b_3(t)$ | $bk$ | ↘ | 0 | ↘ | $-\infty$ / $+\infty$ | ↘ | | $b$ |

**Table 18. Variation table of $b_3(t)$ when $b > 0$, $\beta < 0$, and $k < 0$.**

| $t$ | $-\infty$ | | $\frac{\ln(-\beta)}{b}$ | | $+\infty$ | |
|---|---|---|---|---|---|---|
| $\frac{\partial^2}{\partial t^2}b_3(t)$ | | − | | | + | |
| $\frac{\partial}{\partial t}b_3(t)$ | 0 | ↘ | $-\infty$ | $-\infty$ | ↗ | 0 |
| $b_3(t)$ | $bk$ | ↘ | $-\infty$ | $+\infty$ | ↘ | $b$ |

## 4.4 Fourth form

Let consider:

$$b_4(t) = b\sin(\beta t + \varphi) + k \qquad (40)$$

**Table 19. Variation table of $b_3(t)$ when $b < 0$, $\beta > 0$, and $k > 1$.**

| $t$ | $-\infty$ | | $\frac{\ln \beta}{b}$ | | $+\infty$ |
|---|---|---|---|---|---|
| $\frac{\partial^2}{\partial t^2} b_3(t)$ | | $-$ | $0$ | $+$ | |
| $\frac{\partial}{\partial t} b_3(t)$ | $0$ | ↘ | $\frac{-b^2(k-1)}{4}$ | ↗ | $0$ |
| $b_3(t)$ | $b$ | ↘ | $\frac{b(k+1)}{2}$ | ↘ | $bk$ |

**Table 20. Variation table of $b_3(t)$ when $b < 0$, $\beta > 0$, and $0 < k < 1$.**

| $t$ | $-\infty$ | | $\frac{\ln \beta}{b}$ | | $+\infty$ |
|---|---|---|---|---|---|
| $\frac{\partial^2}{\partial t^2} b_3(t)$ | | $+$ | $0$ | $-$ | |
| $\frac{\partial}{\partial t} b_3(t)$ | $0$ | ↗ | $\frac{-b^2(k-1)}{4}$ | ↘ | $0$ |
| $b_3(t)$ | $b$ | ↗ | $\frac{b(k+1)}{2}$ | ↗ | $bk$ |

**Table 21. Variation table of $b_3(t)$ when $b < 0$, $\beta > 0$, and $-1 < k < 0$.**

| $t$ | $-\infty$ | | $\frac{\ln \beta}{b}$ | | $\frac{\ln(-k\beta)}{b}$ | | $+\infty$ |
|---|---|---|---|---|---|---|---|
| $\frac{\partial^2}{\partial t^2} b_3(t)$ | | $+$ | $0$ | | $-$ | | |
| $\frac{\partial}{\partial t} b_3(t)$ | $0$ | ↗ | $\frac{-b^2(k-1)}{4}$ | | ↘ | | $0$ |
| $b_3(t)$ | $b$ | ↗ | $\frac{b(k+1)}{2}$ | ↗ | $0$ | ↗ | $bk$ |

So

$$\frac{\partial}{\partial t} b_4(t) = b\beta \cos(\beta t + \varphi) \tag{41}$$

And

$$\frac{\partial^2}{\partial t^2} b_4(t) = -b\beta^2 \sin(\beta t + \varphi) \tag{42}$$

**Table 22. Variation table of $b_3(t)$ when $b < 0$, $\beta > 0$, and $k < -1$.**

| $t$ | $-\infty$ | | $\frac{\ln(-k\beta)}{b}$ | | $\frac{\ln \beta}{b}$ | | $+\infty$ |
|---|---|---|---|---|---|---|---|
| $\frac{\partial^2}{\partial t^2} b_3(t)$ | | | $+$ | | $0$ | $-$ | |
| $\frac{\partial}{\partial t} b_3(t)$ | $0$ | | $\nearrow$ | | $\frac{-b^2(k-1)}{4}$ | $\searrow$ | $0$ |
| $b_3(t)$ | $b$ | $\nearrow$ | $0$ | $\nearrow$ | $\frac{b(k+1)}{2}$ | $\nearrow$ | $bk$ |

**Table 23. Variation table of $b_3(t)$ when $b < 0$, $\beta < 0$, and $k > 1$.**

| $t$ | $-\infty$ | | $\frac{\ln(-k\beta)}{b}$ | | $\frac{\ln(-\beta)}{b}$ | | $+\infty$ |
|---|---|---|---|---|---|---|---|
| $\frac{\partial^2}{\partial t^2} b_3(t)$ | | | $+$ | | | $-$ | |
| $\frac{\partial}{\partial t} b_3(t)$ | $0$ | | $\nearrow$ | $+\infty$ | $+\infty$ | $\searrow$ | $0$ |
| $b_3(t)$ | $b$ | $\nearrow$ | $0$ | $\nearrow$ $+\infty$ | $-\infty$ | $\nearrow$ | $bk$ |

**Table 24. Variation table of $b_3(t)$ when $b < 0$, $\beta < 0$, and $0 < k < 1$.**

| $t$ | $-\infty$ | | $\frac{\ln(-\beta)}{b}$ | | $\frac{\ln(-k\beta)}{b}$ | | $+\infty$ |
|---|---|---|---|---|---|---|---|
| $\frac{\partial^2}{\partial t^2} b_3(t)$ | | $-$ | | | | $+$ | |
| $\frac{\partial}{\partial t} b_3(t)$ | $0$ | $\searrow$ | $-\infty$ | $-\infty$ | | $\nearrow$ | $0$ |
| $b_3(t)$ | $b$ | $\searrow$ $-\infty$ | $+\infty$ | $\searrow$ | $+\infty$ $0$ | $\searrow$ | $bk$ |

Special cases of $b_4(t)$ are listed in Table 2. Because $\sin(t)$ is a periodic function, it is necessary to evaluate the rational domain and the codomain of this function. Its period is $2\pi$, so $[-\pi, \pi]$ is considered initially. After it is scaled and shifted horizontally, the domain becomes $[\frac{-\pi-\varphi}{\beta}, \frac{\pi-\varphi}{\beta}]$. After $\sin(t) \in [-1, 1]$ is scaled and shifted vertically, the codomain becomes $[k - |b|, k + |b|]$. $\sin(t)$ and $\cos(t)$ are continuous function, so limitations of $b_4(t)$, $\frac{\partial}{\partial t} b_4(t)$, and $\frac{\partial^2}{\partial t^2} b_4(t)$ at any point

**Table 25. Variation table of $b_3(t)$ when $b < 0$, $\beta < 0$, and $k < 0$.**

| $t$ | $-\infty$ | | $\frac{\ln(-\beta)}{b}$ | | | $+\infty$ | |
|---|---|---|---|---|---|---|---|
| $\frac{\partial^2}{\partial t^2} b_3(t)$ | | $-$ | | | | $+$ | |
| $\frac{\partial}{\partial t} b_3(t)$ | 0 | ↘ | $-\infty$ | $-\infty$ | | ↗ | 0 |
| | | | | $+\infty$ | | | |
| $b_3(t)$ | $b$ | ↘ | $-\infty$ | | $+\infty$ | ↘ | $bk$ |

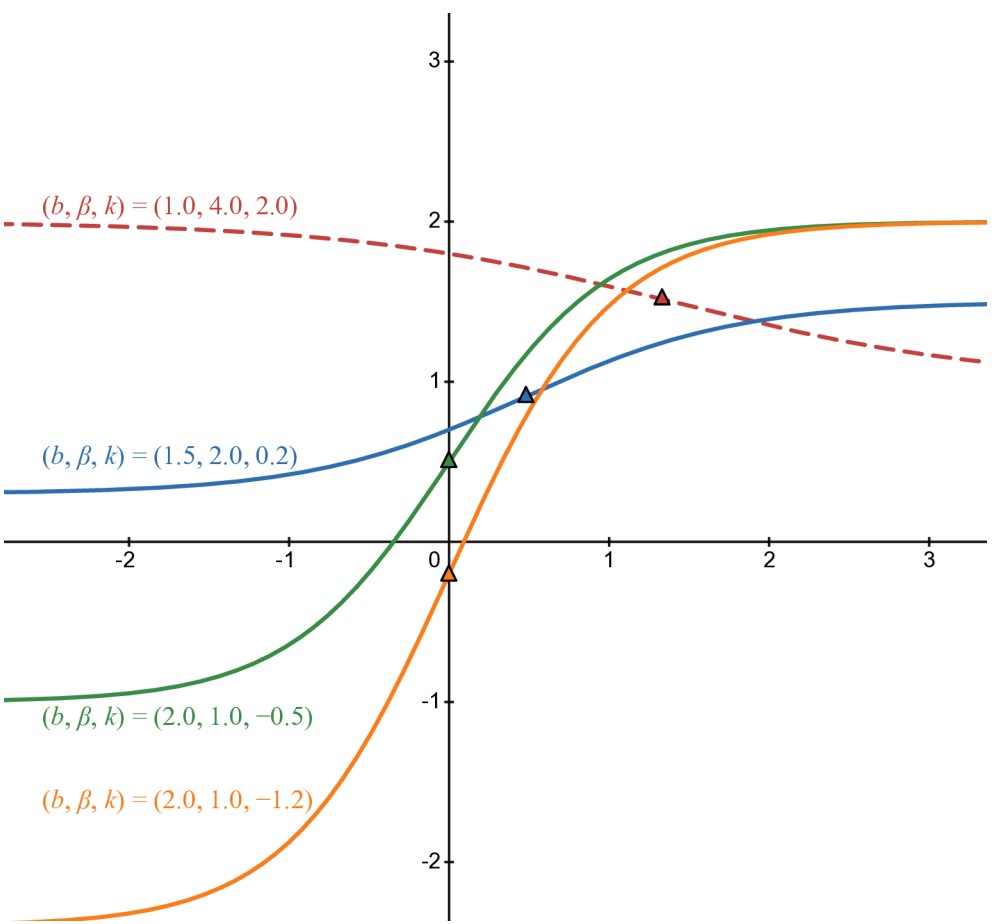

$(b, \beta, k) = (1.0, 4.0, 2.0)$

$(b, \beta, k) = (1.5, 2.0, 0.2)$

$(b, \beta, k) = (2.0, 1.0, -0.5)$

$(b, \beta, k) = (2.0, 1.0, -1.2)$

**Fig 5. Numerical illustration of $b_3(t)$ when $b > 0$ and $\beta > 0$.**

are those values at this point. Let consider:

$$
\begin{aligned}
b\sin[\beta t + \varphi] + k &= (-b)\sin[\beta t + (\varphi + \pi)] + k \\
&= b\sin[(-\beta)t + (-\varphi + \pi)] + k \\
&= (-b)\sin[(-\beta)t + (-\varphi)] + k
\end{aligned}
\tag{43}
$$

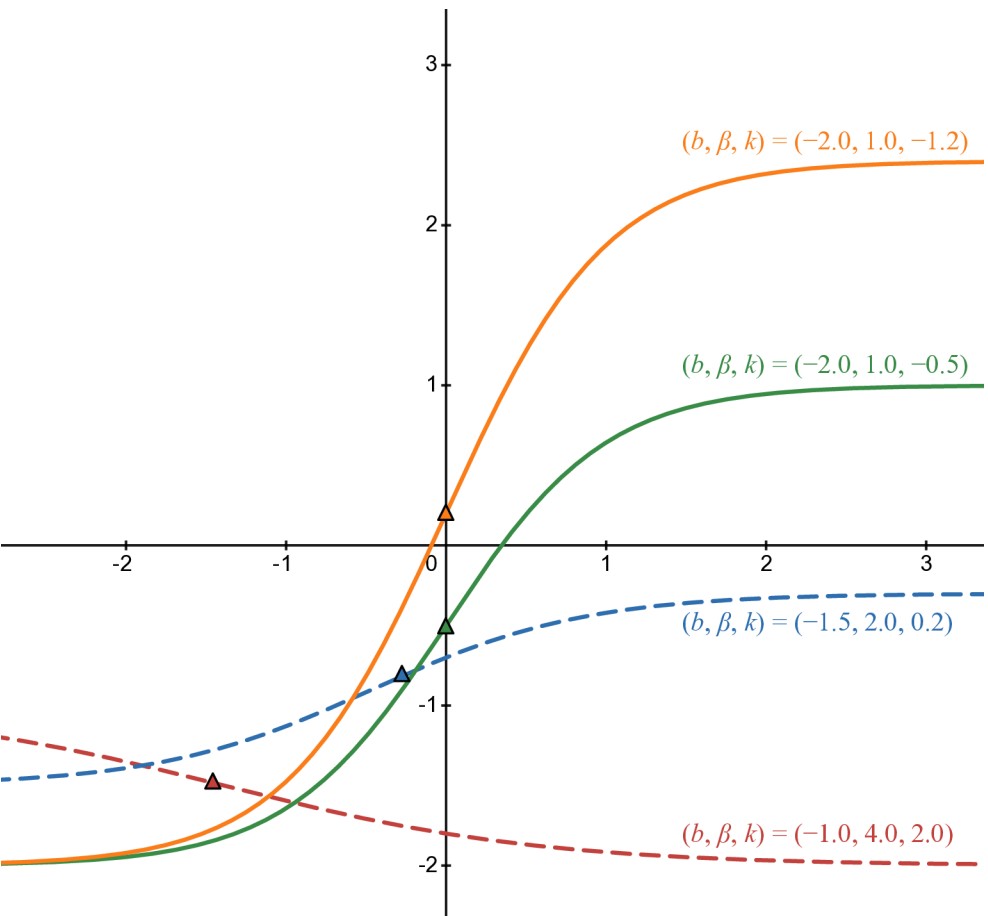

**Fig 6. Numerical illustration of $b_3(t)$ when $b < 0$ and $\beta > 0$.**

Therefore, without loss of generality, let's focus on the case of $(b > 0 \wedge \beta > 0)$ instead of cases of $(b > 0 \wedge \beta < 0)$, $(b < 0 \wedge \beta > 0)$, and $(b < 0 \wedge \beta < 0)$. From those analysis, $b_4(t)$ will be studied in $[\frac{-\pi - \varphi}{\beta}, \frac{\pi - \varphi}{\beta}]$ when $(b > 0 \wedge \beta > 0)$. Let $t^*$ is the solution of $b_4(t) = b\sin(\beta t + \varphi) + k = 0$ in $[0, \frac{\pi - \varphi}{2\beta}]$, i.e. $t^* = [\sin^{-1}(\frac{-k}{b}) - \varphi]\frac{1}{\beta}$. As mentioned before, $b_4(t) \in [k - b, k + b]$, so the range of $b_4(t)$, $\frac{\partial}{\partial t}b_4(t)$, and $\frac{\partial^2}{\partial t^2}b_4(t)$ are:

$$
Y\big(b_4(t)\big) = \begin{cases} Y\big(k - b\big) & \text{if } k > b \wedge k < -b \\ Y\big((t - t^* + 1)(t + t^*)\big) & \text{if } b > k > 0 \\ Y\big((t - t^*)(t + t^* - 1)\big) & \text{if } 0 > k > -b \end{cases} \tag{44}
$$

$$
Y\Big(\frac{\partial}{\partial t}b_4(t)\Big) = Y\Big(-(t - \frac{-\pi - 2\varphi}{2\beta})(t - \frac{\pi - 2\varphi}{2\beta})\Big) \tag{45}
$$

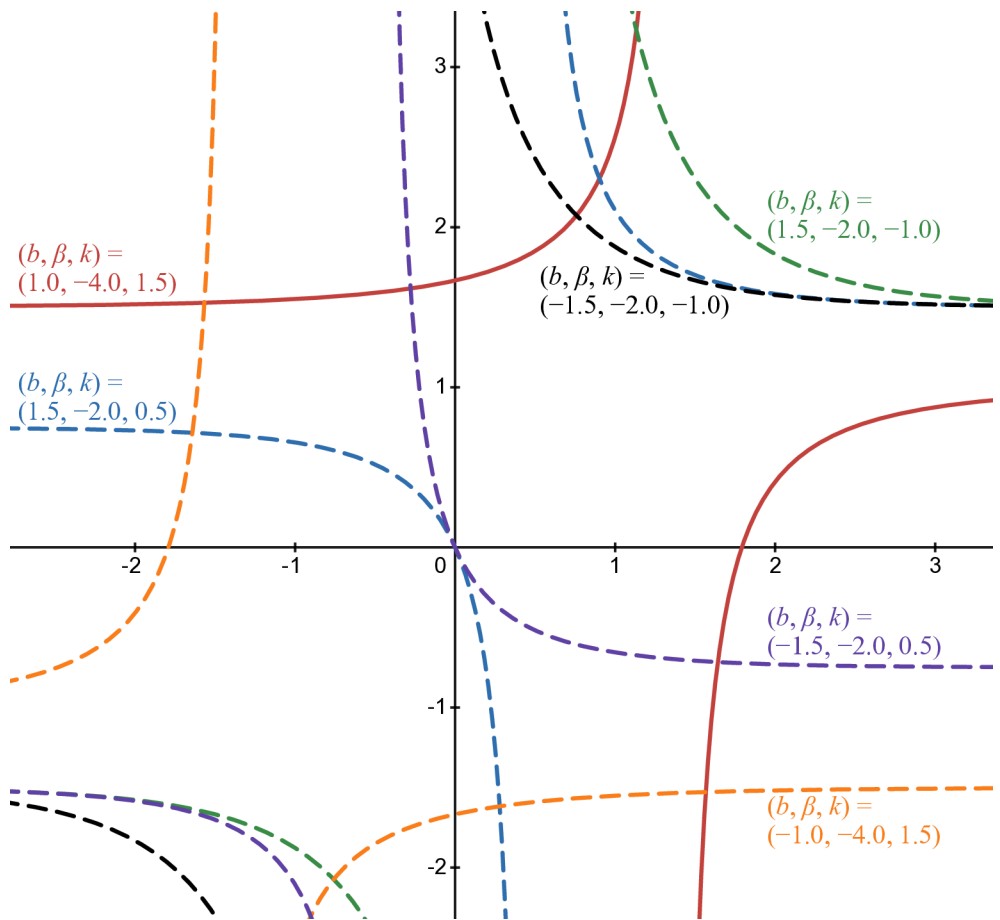

**Fig 7. Numerical illustration of $b_3(t)$ when $\beta < 0$.**

$$\mathrm{Y}\left(\frac{\partial^2}{\partial t^2}b_4(t)\right) = \mathrm{Y}\left(-(t - \frac{-\varphi}{\beta})\right) \tag{46}$$

So $\frac{\partial^2}{\partial t^2}b_4(t) = 0 \Leftrightarrow x_4 \in \{\frac{-\pi-\varphi}{\beta}, \frac{-\varphi}{\beta}, \frac{\pi-\varphi}{\beta}\}$, then inflection points of $b_4(t)$ are $(x_4, y_4) \in \{(\frac{-\pi-\varphi}{\beta}, k), (\frac{-\varphi}{\beta}, k), (\frac{\pi-\varphi}{\beta}, k)\}$. The variation table of $b_4(t)$ are in Tables 26, 27, and 28.

Fig 8 illustrates $b_4(t)$ in some cases of parameters: red line in case of $(b, \beta, \varphi, k) = (2, 0.4, 0, 1)$, blue line in case of $(b, \beta, \varphi, k) = (2, -0.4, -3, 1)$, green line in case of $(b, \beta, \varphi, k) = (-2, 0.4, -3.5, 0.5)$, and orange line in case of $(b, \beta, \varphi, k) = (-2, -0.4, 0.5, -0.5)$. The triangle with the corresponding filled color is the inflection point of each curve.

## 4.5 Rational range of four S-shaped functions

Table 29 summarizes the properties of four S-shaped functions. The first column is the following aspects, which are divided into four groups by thicker lines. ① *Special cases* of parameters of a function take on certain values, and the function could become a simpler form. For example, when $a = 0$, quadratic polynomial $f_2(x) = ax^2 + 2x + 3$ becomes linear polynomial $f_1(x) = 2x + 3$. ② *Domain* is the set of legal inputs of function.

**Table 26. Variation table of $b_4(t)$ when $k > b$ or $k < -b$.**

| $t$ | $\frac{-\pi-\varphi}{\beta}$ | | $\frac{-\pi-2\varphi}{2\beta}$ | | $\frac{-\varphi}{\beta}$ | | $\frac{\pi-2\varphi}{2\beta}$ | | $\frac{\pi-\varphi}{\beta}$ |
|---|---|---|---|---|---|---|---|---|---|
| $\frac{\partial^2}{\partial t^2}b_4(t)$ | 0 | | + | | 0 | | − | | 0 |
| $\frac{\partial}{\partial t}b_4(t)$ | $-b\beta$ | ↗ | 0 | ↗ | $b\beta$ | ↘ | 0 | ↘ | $-b\beta$ |
| $b_4(t)$ | $k$ | ↘ | $k-b$ | ↗ | $k$ | ↗ | $k+b$ | ↘ | $k$ |

**Table 27. Variation table of $b_4(t)$ when $b > k > 0$.**

| $t$ | $\frac{-\pi-\varphi}{\beta}$ | | $t^*-1$ | | $\frac{-\pi-2\varphi}{2\beta}$ | | $-t^*$ | | $\frac{-\varphi}{\beta}$ | | $\frac{\pi-2\varphi}{2\beta}$ | | $\frac{\pi-\varphi}{\beta}$ |
|---|---|---|---|---|---|---|---|---|---|---|---|---|---|
| $\frac{\partial^2}{\partial t^2}b_4(t)$ | 0 | | | | + | | | | 0 | | − | | 0 |
| $\frac{\partial}{\partial t}b_4(t)$ | $-b\beta$ | ↗ | | | 0 | ↗ | | | $b\beta$ | ↘ | 0 | ↘ | $-b\beta$ |
| $b_4(t)$ | $k$ | ↘ | 0 | ↘ | $k-b$ | ↗ | 0 | ↗ | $k$ | ↗ | $k+b$ | ↘ | $k$ |

**Table 28. Variation table of $b_4(t)$ when $0 > k > -b$.**

| $t$ | $\frac{-\pi-\varphi}{\beta}$ | | $\frac{-\pi-2\varphi}{2\beta}$ | | $\frac{-\varphi}{\beta}$ | | $t^*$ | | $\frac{\pi-2\varphi}{2\beta}$ | | $1-t^*$ | | $\frac{\pi-\varphi}{\beta}$ |
|---|---|---|---|---|---|---|---|---|---|---|---|---|---|
| $\frac{\partial^2}{\partial t^2}b_4(t)$ | 0 | | + | | 0 | | | | − | | | | 0 |
| $\frac{\partial}{\partial t}b_4(t)$ | $-b\beta$ | ↗ | 0 | ↗ | $b\beta$ | ↘ | | | 0 | ↘ | | | $-b\beta$ |
| $b_4(t)$ | $k$ | ↘ | $k-b$ | ↗ | $k$ | ↗ | 0 | ↗ | $k+b$ | ↘ | 0 | ↘ | $k$ |

The next aspect to consider is whether the function has an S-shaped pattern or not. As mentioned in Sect 2.2, the 2nd phase, as convex, and the 4th phase, as concave, of the function are often focused on. So ③ *Part S-shaped* is the case where the function has a later part of an S-shaped curve. It is later instead of the earlier part because this part satisfies

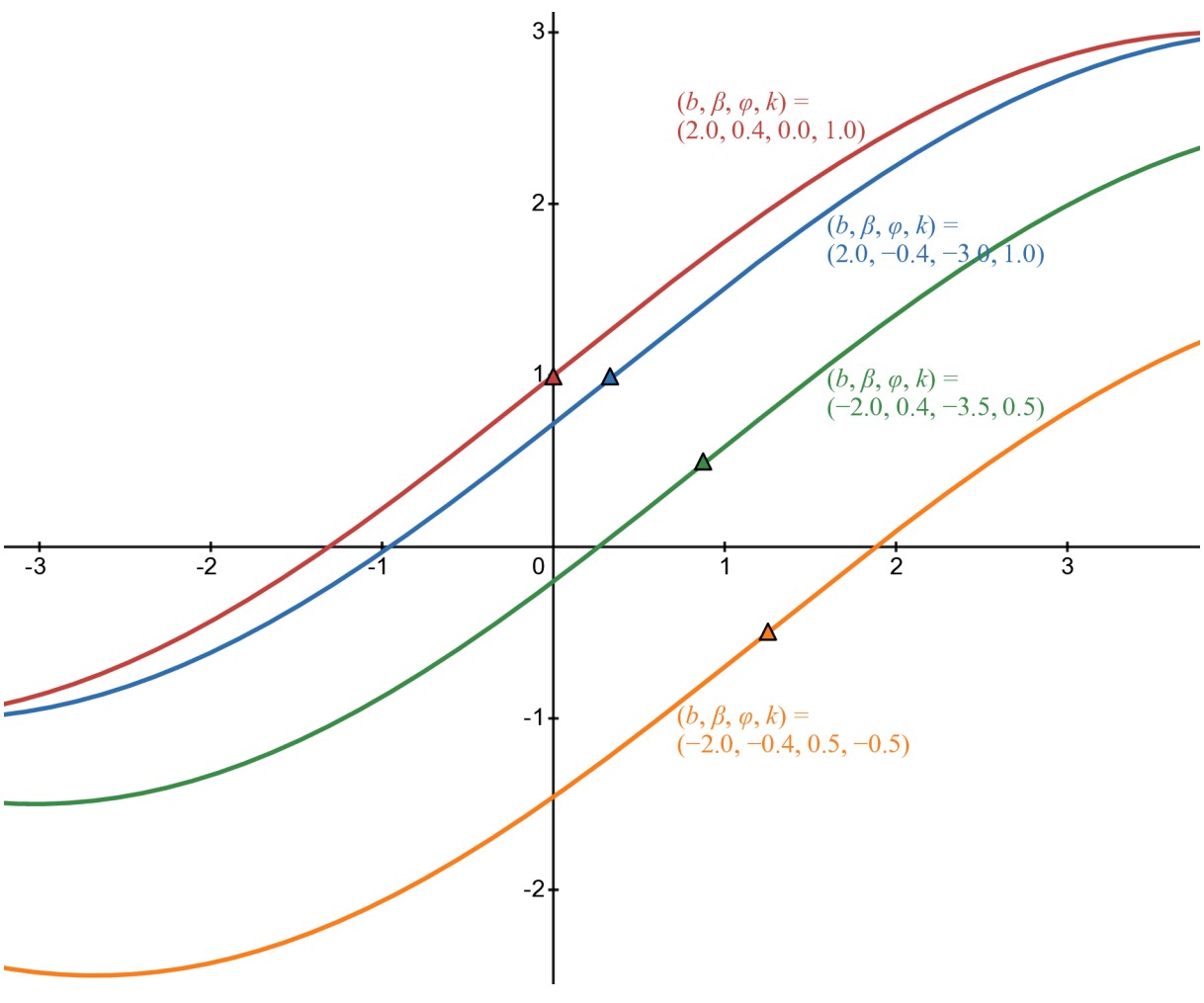

**Fig 8. Numerical illustration of $b_4(t)$.**

the real condition "*the ability of the tester has a limitation, so it converges to a specific value after a long time*". And then, ④ *Fully S-shaped* is when the function has the whole form of S-shaped.

Certainly, the fault detection rate, i.e., the ratio of detected errors to total system errors, has to reach a positive value. So it should analyze whether the S-shaped part of the function is positive or not. ⑤ *Positive part S-shaped* is in case S-shaped part of ③ is positive. ⑥ *Positive 2 phases S-shaped* relates to the newly perspective. As mentioned in Sect 2.2, the 2nd phase of convex and 4th phase of concave are the main parts of an S-shaped curve, so ⑥ evaluates the positive part of the curve that contains both two phases or just one of them. Similar to ⑤, ⑦ *Positive fully S-shaped* is in case the S-shaped curve of ④ is positive.

The next variable is time $t$, which should be analyzed in the positive range. ⑧ *Positive part S-shaped when $t \geq 0$*, ⑨ *Positive 2 phases part S-shaped when $t \geq 0$*, and ⑩ *Positive fully S-shaped when $t \geq 0$* are corresponded cases of ⑤, ⑥, and ⑦ when $t \geq 0$, respectively.

This tabular summary shows the advantages of the transformation of the trigonometric function over the previous S-shaped functions. The first, transformation of a trigonometric function, is the only function that can reach a fully S-shaped

**Table 29**. Rational range of four S-shaped functions.

| | $b_1(t)$ | $b_2(t)$ | $b_3(t)$ | $b_4(t)$ |
|---|---|---|---|---|
| | $b=0 \Rightarrow b_1(t)=0$ | $(b,\beta)=(0,-1) \Rightarrow b_2(t)$ is undefined<br>$b=0 \wedge \beta \neq -1 \Rightarrow b_2(t)=0$ | $(b,\beta)=(0,-1) \Rightarrow b_3(t)$ is undefined<br>$b=0 \wedge \beta \neq -1 \Rightarrow b_3(t)=0$ | As in Table 2 |
| ① | | $\beta=0 \Rightarrow b_2(t)=b$ | $\beta=0 \Rightarrow b_3(t)=b$<br>$k=1 \Rightarrow b_3(t)=b$<br>$k=0 \Rightarrow b_3(t)=b_2(t)$ | |
| ② | $\mathbb{R} \setminus \{\frac{-1}{b}\}$ | $t \in X$ | $t \in X$ | $\mathbb{R}$ |
| ③ | $\forall b$ | $\beta>0$ | $\beta>0 \wedge k<1$<br>$\beta<0 \wedge k>1$ | $\forall b,\beta,\varphi,k$ |
| ④ | ✗<br>$b>0$ | $\beta>0$<br>$b>0 \wedge \beta>0$ | $\beta>0 \wedge k<1$<br>$b>0 \wedge \beta>0 \wedge k<1$ | $\forall b,\beta,\varphi,k$<br>$k+|b|>0$ |
| ⑤ | | | $b>0 \wedge \beta<0 \wedge k>1$<br>$b<0 \wedge \beta>0 \wedge k<0$ | |
| ⑥ | ✗ | $b>0 \wedge \beta>0$ | $b>0 \wedge \beta>0 \wedge -1<k<1$<br>$b<0 \wedge \beta>0 \wedge k<-1$ | $k>0$ |
| ⑦ | ✗ | $b>0 \wedge \beta>0$<br>$b>0 \wedge \beta>0$ | $b>0 \wedge \beta>0 \wedge 0<k<1$<br>$b>0 \wedge \beta>0 \wedge k<1$ | $k-|b|>0$<br>$k+|b|>0$ |
| ⑧ | ✗ | | $b>0 \wedge \beta<0 \wedge k>1$<br>$b<0 \wedge \beta>0 \wedge k<0$ | |
| ⑨ | ✗ | $b>0 \wedge \beta>1$ | $b>0 \wedge \beta>1 \wedge -1<k<1$<br>$b<0 \wedge 1>\beta>0 \wedge k<-1$ | $k>0$ |
| ⑩ | ✗ | ✗ | ✗ | $k-|b|>0$ |

form when $t \geq 0$, which is presented in the last row. The second, this fourth S-shaped form $b_4(t)$ can reach all cases from ③ to ⑨ more easily than other forms. Furthermore, the conditions for achieving them are similar.

## 4.6 Numerical illustration

Fig 9 illustrates the form of 4 S-shaped functions $b_1(t)$ in Eq (7), $b_2(t)$ in Eq (9), $b_3(t)$ in Eq (11), and $b_4(t)$ in Eq (16). The red line is $b_1(t)$ when $b=1$. The blue line is $b_2(t)$ when $b=2$ and $\beta=3$. With $b_3(t)$, the green line is $b=2$, $\beta=3$, and $k=0.2$ while orange line is $b=2$, $\beta=-4$, and $k=1.5$. The purple line is $b_4(t)$ when $b=1.5$, $\beta=-1$, $\varphi=-2.2$, and $k=0.5$.

## 5 Conclusions and future works

This article evaluates the *S-shaped* characteristic of the most famous trigonometric function $\sin(t)$. The transformation of this function is studied to verify that it will be suitable for real conditions. The transformation methods include scaling vertically, scaling horizontally, shifting horizontally, and shifting vertically. The transformed function has been proven to have certain advantages and is suitable for NHPP SRM. The remaining work of SRM introduction is proposing a total fault function $a(t)$ to substitute into Eq (18).

Some long-used S-shaped functions in Eqs (7), (9), and (11) and newly considered function in Eq (16) are analyzed mathematically. The considered characteristics are the domain, codomain, pattern, and the increment process of functions. These mathematical analyses support the derivation of the application domain of the functions, which ensures conformity with real-life conditions and maintains the *S-shaped* property.

In the next works, there are some promising approaches, such as finding $a(t)$ or evaluating some simple forms of $b_4(t)$ by vanishing some parameters or both, to propose a new software reliability model.

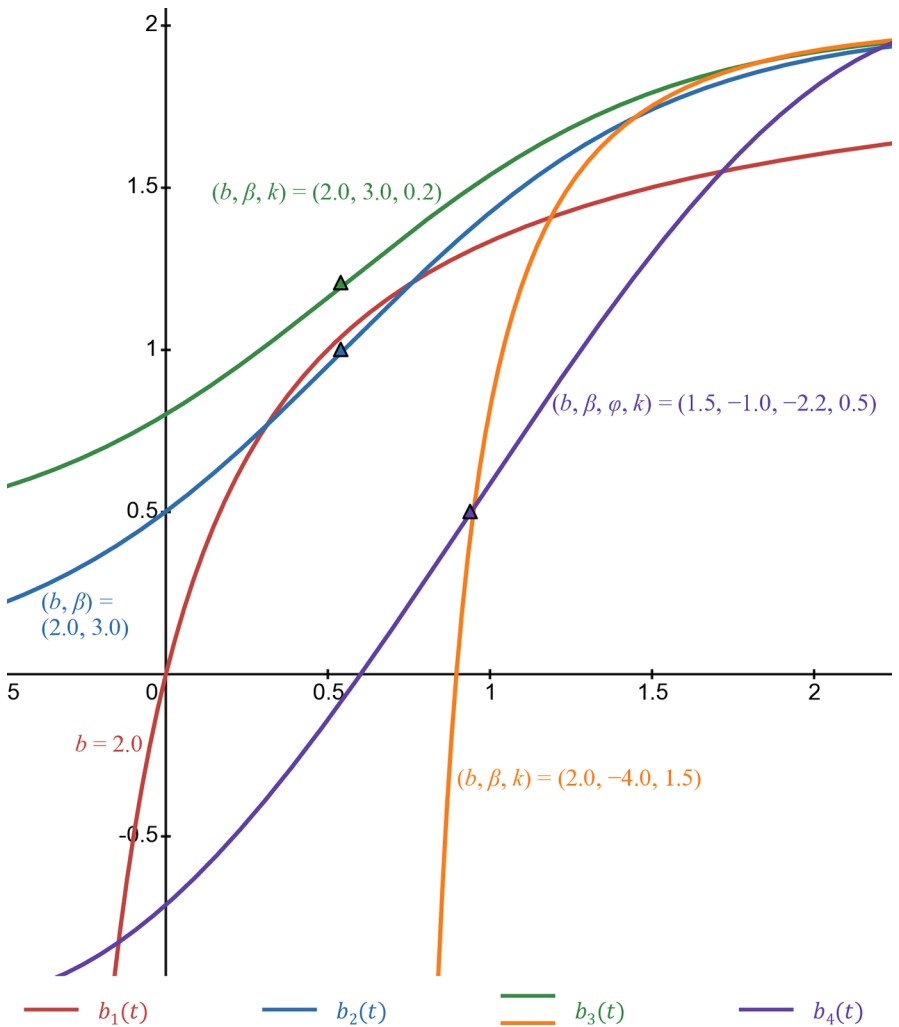

**Fig 9**. **Numerical illustration of 4 types of S-shaped function.**

## Acknowledgments

We also thank Associate Professor Bui Thu-Lam for his careful reading of the manuscript and some propositions for publication.

## Author contributions

**Conceptualization:** Quyet-Thang Huynh, Hung-Cuong Nguyen.

**Formal analysis:** Dai-Nghia Vy, Van-Thuan Nguyen.

**Funding acquisition:** Van-Thuan Nguyen.

**Methodology:** Dai-Nghia Vy, Quyet-Thang Huynh, Hung-Cuong Nguyen.

**Project administration:** Trung-Nghia Phung, Hung-Cuong Nguyen.

**Resources:** Hung-Cuong Nguyen.

**Supervision:** Quyet-Thang Huynh, Trung-Nghia Phung, Hung-Cuong Nguyen.

**Writing – original draft:** Dai-Nghia Vy.

**Writing – review & editing:** Hung-Cuong Nguyen.

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
