## [Decision Letter · Decision Letter 0]

27 Oct 2025

PONE-D-25-53209Trigonometric function transformation and its application in software reliability modelingPLOS ONE

Dear Dr. Nguyen,

Thank you for submitting your manuscript to PLOS ONE. After careful consideration, we feel that it has merit but does not fully meet PLOS ONE’s publication criteria as it currently stands. Therefore, we invite you to submit a revised version of the manuscript that addresses the points raised during the review process.

We look forward to receiving your revised manuscript.

Kind regards,

Mallikarjuna Reddy Kesama, Ph.D.

Academic Editor

PLOS ONE

Journal Requirements:

2. Please note that PLOS One has specific guidelines on code sharing for submissions in which author-generated code underpins the findings in the manuscript. In these cases, we expect all author-generated code to be made available without restrictions upon publication of the work. 

Please review our guidelines at https://journals.plos.org/plosone/s/materials-and-software-sharing#loc-sharing-code and ensure that your code is shared in a way that follows best practice and facilitates reproducibility and reuse.

“This research is supported by Hung Vuong University under grant number HV33.2024”

5. Please note that funding information should not appear in the Acknowledgments section or other areas of your manuscript. We will only publish funding information present in the Funding Statement section of the online submission form. Please remove any funding-related text from the manuscript. 

7.  PLOS requires an ORCID iD for the corresponding author in Editorial Manager on papers submitted after December 6th, 2016. Please ensure that you have an ORCID iD and that it is validated in Editorial Manager. To do this, go to ‘Update my Information’ (in the upper left-hand corner of the main menu), and click on the Fetch/Validate link next to the ORCID field. This will take you to the ORCID site and allow you to create a new iD or authenticate a pre-existing iD in Editorial Manager.

8. Please include a separate caption for each figure in your manuscript.

9. Please upload a new copy of Figures 1 to 9 as the detail is not clear. Please follow the link for more information:  https://journals.plos.org/plosone/s/figures

10. We note you have included a table to which you do not refer in the text of your manuscript. Please ensure that you refer to Table 4 in your text; if accepted, production will need this reference to link the reader to the Table.

Reviewers' comments:

Reviewer's Responses to Questions

**Comments to the Author**

1. Is the manuscript technically sound, and do the data support the conclusions?

Reviewer #1: Yes

Reviewer #2: No

Reviewer #3: Partly

2. Has the statistical analysis been performed appropriately and rigorously?

Reviewer #1: N/A

Reviewer #2: Yes

Reviewer #3: N/A

3. Have the authors made all data underlying the findings in their manuscript fully available?

Reviewer #1: Yes

Reviewer #2: Yes

Reviewer #3: Yes

4. Is the manuscript presented in an intelligible fashion and written in standard English?

Reviewer #1: No

Reviewer #2: Yes

Reviewer #3: No

5. Review Comments to the Author

Please use the space provided to explain your answers to the questions above. You may also include additional comments for the author, including concerns about dual publication, research ethics, or publication ethics. (Please upload your review as an attachment if it exceeds 20,000 characters.)

Reviewer #1: Dear authors,

In this study, the transformation of four types of S-shaped functions using trigonometric functions is demonstrated. The mean value function has also been calculated by considering these functions. The idea of using a transformed function in software reliability is novel, but it needs improvements:

1. The literature review is too brief.

2. Writing improvements are needed: the paper contains some lengthy sentences, and paragraphs are not justified. Condensing and improving logical flow is recommended.

3. It would be better to write the limitations of the transformation, i.e., write the cases when such a transformation is not possible.

4. The representation of Figure 9 may be improved if proper parameters are considered, specially for the red line of b1(t) and orange line of b3(t).

Reviewer #2: Please see the attached file for the details.

The authors propose using the trigonometric sine function transformation to model the S-shaped curve of the cumulative number of detection faults during software testing. But the authors did not verify them through experiments.

Reviewer #3: This article seems to be an extended study of the following manuscript.

"Hung-Cuong, N., Quyet-Thang, H., Trung-Nghia, P., Dai-Nghia, V. (2025). Transformating trigonometric function to apply in software reliability modeling. In: Nghia, P.T., Thai, V.D., Thuy, N.T., Huynh, VN., Van Huan, N. (eds) Advances in Information and Communication Technology. ICTA 2024. Lecture Notes in Networks and Systems, vol 1205. Springer, Cham. https://doi.org/10.1007/978-3-031-80943-9_13".

The novel contribution of this paper is limited.

This paper contains many grammatical mistakes.

English writing need to be improved, severely.

6. PLOS authors have the option to publish the peer review history of their article (what does this mean?). If published, this will include your full peer review and any attached files.

Reviewer #1: No

Reviewer #2: No

Reviewer #3: No

---

## [Author Response · Author response to Decision Letter 1]

15 Nov 2025

Firstly, the authors would like to thank the reviewers for their valuable comments. We answer your question/comment as follows:

A. Academic editor

Comment 1:

Our answer:

We would like to thank you for the comment of the academic editor. We prepare our manuscript using the PLOS ONE style template.

Comment 2:

2. Please note that PLOS One has specific guidelines on code sharing for submissions in which author-generated code underpins the findings in the manuscript. In these cases, we expect all author-generated code to be made available without restrictions upon publication of the work.

Please review our guidelines at https://journals.plos.org/plosone/s/materials-and-software-sharing#loc-sharing-code and ensure that your code is shared in a way that follows best practice and facilitates reproducibility and reuse.

Our answer:

We would like to thank you for the comment of the academic editor. Our manuscript does not have any source code. It provides a theoretical analysis of the application of the transformation of trigonometric functions in Software reliability modelling.

Comment 3:

Our answer:

We would like to thank you for the comment of the academic editor. We prepare our manuscript using the PLOS ONE LaTeX template.

Comment 4:

“This research is supported by Hung Vuong University under grant number HV33.2024”

Our answer:

We would like to thank you for the comment of the academic editor. We have feedback on your comment as follows:

- We move the Role of Funder to the Cover letter as you require.

- Authors Vy Dai-Nghia, Nguyen Van-Thuan, and Nguyen Hung-Cuong are lecturers of Hung Vuong University. This work was done in the lab at our University with the financial support of our University. So we state: "This work was funded by Hung Vuong University (grant HV33.2024)" in the Cover letter.

Comment 5:

5. Please note that funding information should not appear in the Acknowledgments section or other areas of your manuscript. We will only publish funding information present in the Funding Statement section of the online submission form. Please remove any funding-related text from the manuscript.

Our answer:

We would like to thank you for the comment of the academic editor. We will remove the funding information in the Acknowledgments section as you require.

Comment 6:

Our answer:

We would like to thank you for the comment of the academic editor. Our manuscript does not have any source code. It provides a theoretical analysis of the application of the transformation of trigonometric functions in Software reliability modelling.

Comment 7:

7. PLOS requires an ORCID iD for the corresponding author in Editorial Manager on papers submitted after December 6th, 2016. Please ensure that you have an ORCID iD and that it is validated in Editorial Manager. To do this, go to ‘Update my Information’ (in the upper left-hand corner of the main menu), and click on the Fetch/Validate link next to the ORCID field. This will take you to the ORCID site and allow you to create a new iD or authenticate a pre-existing iD in Editorial Manager.

Our answer:

We would like to thank you for the comment of the academic editor. I updated my ORCID in My Information in the Editorial Manager system. My ORCID is 0009-0002-5974-8518.

Comment 8:

8. Please include a separate caption for each figure in your manuscript.

Our answer:

We would like to thank you for the comment of the academic editor. We provide a separate caption for each figure in our manuscript.

Comment 9:

9. Please upload a new copy of Figures 1 to 9 as the detail is not clear. Please follow the link for more information: https://journals.plos.org/plosone/s/figures

Our answer:

We would like to thank you for the comment of the academic editor. We update the new version of figures in .tiff and .eps format to improve the quality of figures.

Comment 10:

10. We note you have included a table to which you do not refer in the text of your manuscript. Please ensure that you refer to Table 4 in your text; if accepted, production will need this reference to link the reader to the Table.

Our answer:

We would like to thank you for the comment of the academic editor. It is our mistake. We must refer to Tables 3 and 4 instead of Tables 2 and 3. We update the reference of Table 4 as highlighted on page 10.

Comment 11:

Our answer:

We would like to thank you for the comment of the academic editor. Reviewer 3 mentioned that this work is extended from "Hung-Cuong, N., Quyet-Thang, H., Trung-Nghia, P., Dai-Nghia, V. (2025). Transformating trigonometric function to apply in software reliability modeling. In: Nghia, P.T., Thai, V.D., Thuy, N.T., Huynh, VN., Van Huan, N. (eds) Advances in Information and Communication Technology. ICTA 2024. Lecture Notes in Networks and Systems, vol 1205. Springer, Cham. https://doi.org/10.1007/978-3-031-80943-9_13". So we cite this paper in subsection 3.2 on page 4 to inherit this previous work.

B. Reviewer #1

Comment 1:

Dear authors,

In this study, the transformation of four types of S-shaped functions using trigonometric functions is demonstrated. The mean value function has also been calculated by considering these functions. The idea of using a transformed function in software reliability is novel, but it needs improvements:

1. The literature review is too brief.

Our answer:

We would like to thank you for the comment of reviewer. We add Table 1 on page 4 to list some studied NHPP SRM that use the S-shaped function. Those models provide an overview of previous works.

Comment 2:

2. Writing improvements are needed: the paper contains some lengthy sentences, and paragraphs are not justified. Condensing and improving logical flow is recommended.

Our answer:

We would like to thank you for the comment of reviewer. We have checked the grammatical mistakes in English and edited our manuscript carefully.

Comment 3:

3. It would be better to write the limitations of the transformation, i.e., write the cases when such a transformation is not possible.

Our answer:

We would like to thank you for the comment of reviewer. In our work, the most famous trigonometric function, sin(t), is studied. The domain of this function is the full set of real numbers, i.e., t ∈ R. So this function can be transformed in all cases.

Comment 4:

4. The representation of Figure 9 may be improved if proper parameters are considered, specially for the red line of b1(t) and orange line of b3(t).

Our answer:

We would like to thank you for the comment of reviewer. We have feedback on your comment as follows:

As mentioned on page 10: "We can see that this form is not fully S-shaped because it only contains the two last phases of the S-shaped curve as described in sub-section 2.2". Furthermore, b1(t) must pass through the origin of the Cartesian coordinate system, i.e., t = 0, then b1(t) = 0. So b1(t) is illustrated by b = 2.0.

- If β < 0, b3(t) has the similar shape with b1(t) which is "not fully S-shaped". So b3(t) is illustrated by (b, β, k) = (2.0, −4.0, 1.5).

So we propose to retain this figure to illustrate the S-shape pattern of the 4 types of S-shaped functions.

C. Reviewer #2

Comment 1:

The authors propose using the trigonometric sine function transformation to model the S-shaped curve of the cumulative number of detection faults during software testing. But the authors did not verify them through experiments.

Our answer:

We would like to thank you for the comment of reviewer. Our manuscript provides a theoretical analysis of the application of the transformation of trigonometric functions in Software reliability modelling without any experimental evaluation. The experimental results will be studied and discussed in future works.

Comment 2:

The problems of this paper are as follows,

1) Although the authors introduced the property of trigonometric sine function transformation, such as bsin(t) and b sin(βt + φ) + k, they indeed ignored the limitation of t and t ≥ 0 during software testing. This restriction will seriously weaken the accuracy of the fault detection rate based on the sine function. I suggest the author compare the fitting and predictive performance of the model proposed in Table 1 with other models. In addition, previous studies have used the sine function to establish software reliability models. I suggest the author to include those models for comparison when conducting experiments.

Our answer:

We would like to thank you for the comment of reviewer. We have feedback on your comment as follows:

- In this work, we evaluate the mathematical properties of 4 types of S-shaped functions. All of the previous types of S-shaped function, i.e., b1(t), b2(t), and b3(t), are not considered in the case of t < 0.

- Our manuscript provides a theoretical analysis of the application of the transformation of trigonometric functions in Software reliability modelling without any experimental evaluation. The experimental results will be studied and discussed in future works.

- We do not know any SRM that uses the trigonometric sine function. If there is any model that uses the trigonometric sine function, we will use it to compare with our proposed models in our next work.

Comment 3:

2) In Subsection 2.2, “Phase 3. The function goes through inflection point, which is the point that the derivation of the function changes from positive to negative quantities.” I disagree with this statement. The S-shaped function (i.e. mean value function) in the software reliability model is a non-decreasing function in any situation. In other words, the derivative of the S-shaped function (mean value function) in the software reliability model is never negative under any circumstances.

Our answer:

We would like to thank you for the comment of reviewer. We agree with your statement: "The S-shaped function (i.e. mean value function) in the software reliability model is a non-decreasing function in any situation.". However, subsection 2.2 discusses the value of the derivation of the function. For example, b2(t) has form:

b_2 (t)=b 1/(1+βe^(-bt) )

And the first derivation of the function:

∂/∂t b_2 (t)=(b^2 βe^(-bt))/(1+βe^(-bt) )^2

And in subsection 4.2, we present that the inflection point of b2(t) is (ln⁡β/b, b/2). This means that when t<ln⁡β/b, we have ∂/∂t b_2 (t)>0. And vice versa, when t>ln⁡β/b, we have ∂/∂t b_2 (t)<0. Furthermore, we change from "the derivation of the function" to "the first derivation of the function" on page 3 to emphasize our opinion.

Comment 4:

3) Some grammar errors are as follows, Under equations (7,9,11), “Subtitute...” should be “Substitute...”, you must check it in the whole paper. In the last paragraph of Subsection 3.1, “sin(t) increase...” should be “sin(t) increases...”.

Our answer:

We would like to thank you for the comment of reviewer. We have checked the grammatical mistakes in English and edited our manuscript carefully.

D. Reviewer #3

Comment 1:

This article seems to be an extended study of the following manuscript.

"Hung-Cuong, N., Quyet-Thang, H., Trung-Nghia, P., Dai-Nghia, V. (2025). Transformating trigonometric function to apply in software reliability modeling. In: Nghia, P.T., Thai, V.D., Thuy, N.T., Huynh, VN., Van Huan, N. (eds) Advances in Information and Communication Technology. ICTA 2024. Lecture Notes in Networks and Systems, vol 1205. Springer, Cham. https://doi.org/10.1007/978-3-031-80943-9_13".

Our answer:

We would like to thank you for the comment of reviewer. So we cite this paper in subsection 3.2 on page 4 to inherit this previous work.

Comment 2:

The novel contribution of this paper is limited.

Our answer:

We would like to thank you for the comment of reviewer. We have feedback on your comment as follows. In abstract, we address our contribution in this manuscript as follows: "The first contribution is a deep mathematical appreciation of three well-known S-shaped functions. The second is the mathematical transformation of the trigonometric function to meet the real assumption. The last are the advantages and the applicability of this transformation in software reliability modeling.". All of those contributions have not been studied in any previous works.

Comment 3:

This paper contains many grammatical mistakes.

English writing need to be improved, severely.

Our answer:

We would like to thank you for the comment of reviewer. We have checked the grammatical mistakes in English and edited our manuscript carefully.

---

## [Decision Letter · Decision Letter 1]

4 Dec 2025

Trigonometric function transformation and its application in software reliability modeling

PONE-D-25-53209R1

Dear Dr. Van Thuan Nguyen,

We’re pleased to inform you that your manuscript has been judged scientifically suitable for publication and will be formally accepted for publication once it meets all outstanding technical requirements.

Kind regards,

Mallikarjuna Reddy Kesama, Ph.D.

Academic Editor

PLOS One

Additional Editor Comments (optional):

accept

Reviewers' comments:

Reviewer's Responses to Questions

**Comments to the Author**

1. If the authors have adequately addressed your comments raised in a previous round of review and you feel that this manuscript is now acceptable for publication, you may indicate that here to bypass the “Comments to the Author” section, enter your conflict of interest statement in the “Confidential to Editor” section, and submit your "Accept" recommendation.

Reviewer #1: All comments have been addressed

2. Is the manuscript technically sound, and do the data support the conclusions?

Reviewer #1: Yes

3. Has the statistical analysis been performed appropriately and rigorously?

Reviewer #1: Yes

4. Have the authors made all data underlying the findings in their manuscript fully available?

Reviewer #1: Yes

5. Is the manuscript presented in an intelligible fashion and written in standard English?

Reviewer #1: Yes

6. Review Comments to the Author

Reviewer #1: Since the authors answered all my queries, the article may be accepted for publication after checking the format.

7. PLOS authors have the option to publish the peer review history of their article (what does this mean?). If published, this will include your full peer review and any attached files.

Reviewer #1: No

<qb-div data-qb-element="re-enable-flow" style="z-index: 2147483647; max-width: 1px; max-height: 1px; box-sizing: border-box; position: fixed; top: 10px; right: 10px;"><qb-div style="all: initial !important;"></qb-div></qb-div>

---

## [Editor Report · Acceptance letter]

PONE-D-25-53209R1

PLOS One

Dear Dr. Nguyen,

I'm pleased to inform you that your manuscript has been deemed suitable for publication in PLOS One. Congratulations! Your manuscript is now being handed over to our production team.

Kind regards,

on behalf of

Dr. Mallikarjuna Reddy Kesama

Academic Editor

PLOS One